# UNIFIED MIRROR DESCENT:
# TOWARDS A BIG UNIFICATION OF DECISION MAKING

## ABSTRACT

Decision-making problems, encompassing single-agent, cooperative multi-agent, competitive multi-agent, and mixed cooperative-competitive cases, are ubiquitous in real-world applications. In the past several decades, substantial strides in theoretical and algorithmic advancements have been achieved within these fields. Nevertheless, these fields have been predominantly evolving independently, giving rise to a fundamental question: *Can we develop a single algorithm to effectively tackle all these scenarios?* In this work, we embark upon an exploration of this question by introducing a unified approach to address all types of decision-making scenarios. First, we propose a unified mirror descent (UMD) algorithm which synergistically integrates multiple base policy update rules. Specifically, at each iteration, the new policy of an agent is computed by weighting the base policies obtained through different policy update rules. One of the advantages of UMD is that only minimal modifications are required when integrating new policy update rules. Second, as the evaluation metric of the resulting policy is non-differentiable with respect to the weights of the base policies, we propose a simple yet effective zero-order method to optimize these weights. Finally, we conduct extensive experiments on 24 benchmark environments, which shows that in over 87% (21/24) games UMD performs better than or on-par with the base policies, demonstrating its potential to serve as a unified approach for various decision-making problems. To our knowledge, this is the first attempt to comprehensively study all types of decision-making problems under a single algorithmic framework.

## 1 INTRODUCTION

Decision-making problems spanning from single-agent to multi-agent settings are ubiquitous in our daily life (Rizk et al., 2018). In single-agent contexts, reinforcement learning (RL) has proved effective in real-world applications ranging from robotic navigation (Singh et al., 2022) to plasma control in nuclear fusion research (Degrave et al., 2022), and substantial progress on theoretical underpinnings of policy optimization has been made in recent works (Mei et al., 2020; Zhan et al., 2023; Gaur et al., 2023). Moving beyond single-agent RL, the challenge inherently becomes more intricate, and various methods have been tailored to effectively tackle different multi-agent problems, especially for multi-agent cooperative RL (Lowe et al., 2017; Foerster et al., 2018; Rashid et al., 2018; Son et al., 2019; Wang et al., 2021) and zero-sum games (Bailey & Piliouras, 2018; Kangarshahi et al., 2018; Wibisono et al., 2022; Kozuno et al., 2021; Lee et al., 2021; Jain et al., 2022; Ao et al., 2023; Liu et al., 2023; Cen et al., 2023; Sokota et al., 2023). Nevertheless, these fields have been predominantly evolving independently. Furthermore, it remains elusive and unexplored when venturing to more complicated general-sum cases (Song et al., 2022) where the sum of agents' payoffs is non-zero and mixed cooperative-competitive cases (Xu et al., 2023) where agents in the same team need to cooperate with each other. This motivates us to answer a fundamental question:

> *Can we leverage **a single reinforcement learning algorithm with minimal modifications** to handle the decision-making of single-agent, cooperative multi-agent, competitive multi-agent, and mixed cooperative-competitive cases?*

As one of the most popular algorithms, mirror descent (MD) (Vural et al., 2022) has demonstrated its power in RL (Tomar et al., 2022) and game theory (Cen et al., 2023; Sokota et al., 2023). With

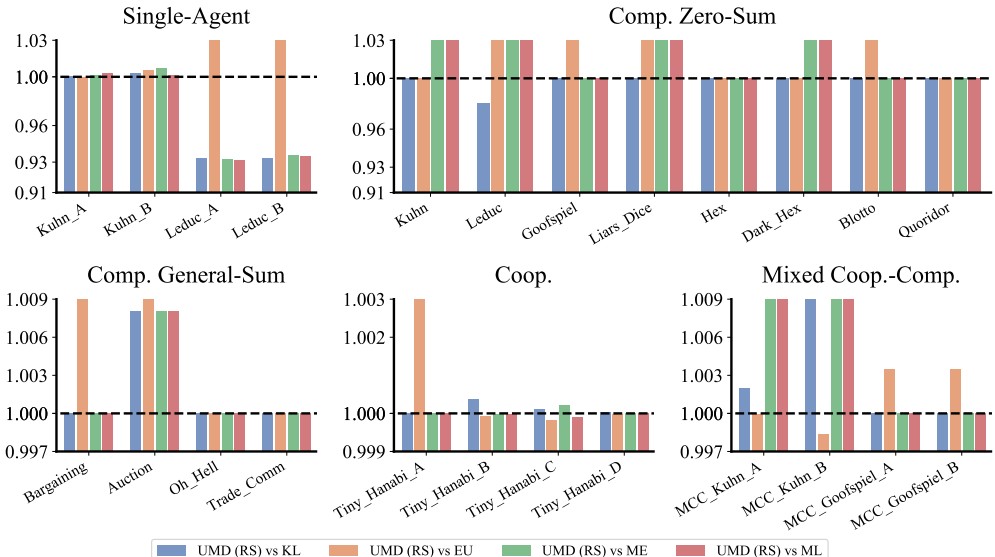

Figure 1: The Y-axis is the normalized improvement of UMD (RS) versus baselines: $> 1$ means UMD (RS) outperforms the baselines, $= 1$ means UMD (RS) matches the baselines, and $< 1$ means UMD (RS) lags behind the baselines. (i) In over 87% (21/24) games UMD (RS) outperforms or matches the baselines. (ii) The numbers of games in which UMD (RS) significantly outperforms the baselines are: 4 (KL), 11 (EU), 7 (ME), and 7 (ML). (iii) For the four baselines, none of them can consistently outperform all the others across all types of decision-making problems.

different mirror maps such as the negative entropy and Euclidean norm, various policy update rules have been induced in the literature. Despite their success in either theoretical convergence guarantee or strong empirical performance, they are typically limited to single-agent RL (Tomar et al., 2022; Zhan et al., 2023; Gaur et al., 2023) and zero-sum games (Bailey & Piliouras, 2018; Kangarshahi et al., 2018; Wibisono et al., 2022; Kozuno et al., 2021; Lee et al., 2021; Jain et al., 2022; Ao et al., 2023; Liu et al., 2023; Cen et al., 2023; Sokota et al., 2023). For general-sum (Bai et al., 2021; Song et al., 2022) and mixed cooperative-competitive settings (Kurach et al., 2020; Xu et al., 2023), the most straightforward idea is to directly apply contemporary MD methods to solve these more complicated scenarios. However, there is no affirmative answer to the question of which one can consistently outperform all the others when applying these MD methods to different decision-making problems. Even under the tabular setting, a comprehensive empirical study of the performance of contemporary MD methods in various types of decision-making problems is lacking.

In this work, we aim to develop *a single reinforcement learning algorithm* which will be individually adopted by each agent (i.e., decentralized execution) while still effectively handling different types of decision-making problems. As this is the first attempt, we focus on the tabular setting, which, though has been often studied in single-agent and zero-sum games, yet unexplored for more complicated general-sum and mixed cooperative-competitive settings. Our contributions are threefold.

- We propose a unified mirror descent (UMD) algorithm by synergistically integrating multiple policy update rules induced by different mirror maps (e.g., negative entropy and Euclidean norm). More specifically, at each iteration, the new policy of an agent is computed by weighting the base policies derived from the policy update rules. UMD is easy to extend to integrate new policy update rules with only minimal modifications required.

- Optimizing the weights assigned to different base policies, unfortunately, is non-trivial as the evaluation metric of the resulting policy (e.g., the return in single-agent settings) is non-differentiable with respect to these weights. To address this issue, we propose a simple yet effective zero-order hyperparameter optimization (HPO) method to optimize these weights. Different from existing zero-order HPO methods, the performance improvement is used to only determine the update direction of the weights rather than the update magnitude, which is more effective when the evaluation metric converges relatively fast.

- We conduct extensive experiments on 24 benchmark games which are divided into 5 types (Figure 1): single-agent, competitive zero-sum, competitive general-sum, cooperative, and mixed cooperative-competitive. Experimental results show that in over 87% (21/24) games UMD performs better than or on-par with all the base policies, demonstrating its potential to serve as a unified approach for a wide range of decision-making problems. Moreover, to our knowledge, our experiments also provide the first comprehensive empirical study of all types of (tabular) decision-making problems under a single algorithmic framework.

## 2 RELATED WORK

Mirror descent (MD) (Vural et al., 2022) has demonstrated effectiveness in learning optimal policies in single-agent RL (Tomar et al., 2022) and proved the last-iterate convergence in learning approximate equilibrium in zero-sum games (Bailey & Piliouras, 2018; Kangarshahi et al., 2018; Wibisono et al., 2022; Kozuno et al., 2021; Lee et al., 2021; Jain et al., 2022; Ao et al., 2023; Liu et al., 2023; Cen et al., 2023; Sokota et al., 2023). Moving beyond zero-sum games, the last-iterate convergence of MD was established for several classes of games such as polymatrix and potential games (Anagnostides et al., 2022). In this work, instead of theoretically comparing the policy update rules induced by different mirror maps which could be difficult, particularly for general-sum (Bai et al., 2021; Song et al., 2022) and mixed cooperative-competitive cases (Kurach et al., 2020; Xu et al., 2023), we propose a unified mirror descent (UMD) algorithm which generalizes multiple policy update rules. UMD is easy to extend to integrate new policy update rules with minimal modifications required. Moreover, our experiments also provide the first comprehensive study of all types of (tabular) decision-making problems under a single algorithmic framework.

Our work is also related to zero-order hyperparameter optimization (HPO) which can update the parameters of interest without access to the true gradient, which has been extensively adopted in adversarial robustness of deep neural networks (Ilyas et al., 2018), meta-learning (Song et al., 2020), and transfer learning (Tsai et al., 2020). The most related work is (Wang et al., 2022), which applied zero-order optimization methods to neural architecture search (NSA) and established the connection between gradient-based NAS and zero-order methods. In this work, we propose a simple yet effective zero-order HPO method in which the performance improvement is used to only determine the update direction of the weights rather than the update magnitude, which is more effective than existing methods in (Wang et al., 2022) when the evaluation metric converges relatively fast.

## 3 PROBLEM STATEMENT

A decision-making problem, either single-agent, cooperative multi-agent, competitive multi-agent, or mixed cooperative-competitive settings, can be described as a decentralized partially observable Markov decision process (Dec-POMDP) (Oliehoek & Amato, 2016) formulated as a tuple $\langle \mathcal{N}, \mathcal{S}, \mathcal{A}, \mathcal{O}, \Omega, P, R, \gamma \rangle$. $\mathcal{N}$ is the set of agents. $\mathcal{S}$ is the (finite) set of the states. $\mathcal{A} = \times_{i \in \mathcal{N}} \mathcal{A}_i$ and $\mathcal{O} = \times_{i \in \mathcal{N}} \mathcal{O}_i$ where $\mathcal{A}_i$ and $\mathcal{O}_i$ are the (finite) set of actions and observations of agent $i$, respectively. We denote $\boldsymbol{a} \in \mathcal{A}$ as the joint action of agents where $a_i \in \mathcal{A}_i$ is the action of agent $i$. $\Omega = \times_{i \in \mathcal{N}} \Omega_i$ where $\Omega_i : \mathcal{S} \times \mathcal{A} \to \mathcal{O}_i$ is the observation function, which specifies the observation $o_i \in \mathcal{O}_i$ of agent $i$ when agents take $\boldsymbol{a} \in \mathcal{A}$ at the state $s \in \mathcal{S}$. $P : \mathcal{S} \times \mathcal{A} \times \mathcal{S} \to [0, 1]$ is the transition function which specifies the probability of transiting to $s' \in \mathcal{S}$ when agents take $\boldsymbol{a} \in \mathcal{A}$ at the state $s \in \mathcal{S}$. $R = \{r_i\}_{i \in \mathcal{N}}$ where $r_i : \mathcal{S} \times \mathcal{A} \to \mathbb{R}$ is the reward function of agent $i$ and $\gamma \in [0, 1)$ is the discount factor. At time step $t \geq 0$, each agent has an action-observation history (i.e., a decision point) $\tau_i^t \in \mathcal{T}_i$ where $\mathcal{T}_i = (\mathcal{O}_i \times \mathcal{A}_i)^t$ and constructs its individual policy $\pi_i : \mathcal{T}_i \times \mathcal{A}_i \to [0, 1]$ to maximize its own return. The joint policy of agents is denoted as $\boldsymbol{\pi} = (\pi_i)_{i \in \mathcal{N}}$. Then, the value function of agent $i$ is defined as $V_i(\boldsymbol{\pi}) = \mathbb{E}[\sum_{t=0}^{\infty} \gamma^t r_i^t | s_0, \boldsymbol{\pi}]$ where $r_i^t$ is the agent $i$'s reward at time step $t$ and $s_0$ is the initial state. Moreover, at decision point $\tau_i^t$, the action-value function of an action $a \in \mathcal{A}_i$ is defined as $Q(\tau_i^t, a, \boldsymbol{\pi}) = \mathbb{E}[\sum_{h=t+1}^{\infty} \gamma^h r_i^h | \tau_i^t, a_i^t = a, \boldsymbol{\pi}]$.

We first introduce the solution concepts used in this work. A policy $\pi_i$ of agent $i$ is said to be optimal[1] if it is optimal in every decision point belonging to the agent. In single-agent and cooperative settings, this optimal policy achieves the maximum return for the agent/team. In (multi-agent) competitive and mixed cooperative-competitive settings, we use Nash equilibrium (NE) as the solution

---

[1]Precisely, it is soft optimal (Sokota et al., 2023). We omit the prefix *soft* for brevity.

concept. A joint policy is an NE if each agent's policy is optimal, given that other agents do not change their policies. Formally, let $\boldsymbol{\pi}^* = (\pi_i^*)_{i \in \mathcal{N}}$ be the NE. Then, agent $i$'s policy satisfies:

$$\pi_i^*(\tau_i^t) = \arg\max_{\pi_i \in \Pi_i} \mathbb{E}_{a \sim \pi_i(\tau_i^t)} Q(\tau_i^t, a, \{\pi_i, \boldsymbol{\pi}_{-i}^*\}) + \epsilon \mathcal{H}(\pi_i), \quad \forall \tau_i^t, \tag{1}$$

where $\Pi_i = \Delta(\mathcal{A}_i)$ is agent $i$'s policy space and $\Delta(\cdot)$ is the action simplex, $\boldsymbol{\pi}_{-i}^*$ denote the joint policy of all agents except agent $i$, $\epsilon$ is the regularization temperature, and $\mathcal{H}$ is Shannon entropy.

In single-agent and cooperative settings, the evaluation metric for a policy/joint policy is the expected return of the agent/team. In other cases, the evaluation metric for a joint policy is the distance of the policy to the NE, called the NE-Gap. Formally, the NE-Gap of the joint policy $\boldsymbol{\pi}$ is defined as NE-Gap$(\boldsymbol{\pi}) = \sum_{i \in \mathcal{N}} [V_i(\pi_i^{\mathrm{BR}}, \boldsymbol{\pi}_{-i}) - V_i(\boldsymbol{\pi})]$ where $\pi_i^{\mathrm{BR}}$ is the best response (BR) policy of agent $i$ against other agents. Note that in mixed cooperative-competitive cases, the BR policy should be the team's BR policy (see Appendix C.2 for more details on the evaluation protocol).

Many methods have been developed to solve the problem (1) for single-agent (Tomar et al., 2022) and multi-agent settings (Sokota et al., 2023). However, for multi-agent settings, most of the existing works typically focus on two-player zero-sum games, while little has been known for more complicated cases including general-sum and mixed cooperative-competitive settings. Nevertheless, notice that Eq. (1) provides a unified description for all the decision-making scenarios as it presents the optimality condition from a single agent's perspective. This motivates us to develop a unified policy update rule, which, when individually adopted by each agent, offers an efficient method to solve the problem (1), i.e., achieving optimal expected return in single-agent and cooperative settings while finding approximate NE in competitive and mixed cooperative-competitive cases.

## 4 UNIFIED MIRROR DESCENT

As we aim to develop a unified policy update rule that will be individually adopted by each agent in each decision point, we only focus on the policy learning of agent $i$ in a single decision point $\tau_i \in \mathcal{T}_i$ and henceforth, the index $i$ and $\tau_i$ are ignored as they are clear from the context, and with a slight abuse of notation, we use $\mathcal{A}$ to represent the action set $\mathcal{A}_i$ of agent $i$. Let $\pi \in \Pi$ be the agent's policy and $Q(a)$ be the action-value of an action $a \in \mathcal{A}$. Note that the joint policy of other agents $\boldsymbol{\pi}_{-i}$ is also omitted in the action-value function. Then, we aim to solve the following problem:

$$\pi^* = \arg\max_{\pi \in \Pi} \mathbb{E}_{a \sim \pi} Q(a) + \epsilon \mathcal{H}(\pi). \tag{2}$$

In single-agent and two-player zero-sum (i.e., purely competitive) settings, the most commonly used method to solve the problem (2) is mirror descent. Formally, the update rule takes the form

$$\pi_{k+1} = \arg\max_{\pi \in \Pi} \mathbb{E}_{a \sim \pi} Q_k(a) - f(\pi, \pi_k), \tag{3}$$

where $k \leq K$ is the iteration, $Q_k$ is the action-value function induced by $\pi_k$, $f$ is called the regularizer. As each choice of $f$ induces a specific policy update rule, in Section 4.1, we present four candidates and then propose a new update rule by integrating them with minimal modifications.

### 4.1 A UNIFIED POLICY UPDATE RULE

Let $f(\pi, \pi_k) = \epsilon \mathcal{B}_\phi(\pi, \rho) + \frac{1}{\eta} \mathcal{B}_\phi(\pi, \pi_k)$. Then, we have

$$\pi_{k+1} = \arg\max_{\pi \in \Pi} \mathbb{E}_{a \sim \pi} Q_k(a) - \epsilon \mathcal{B}_\phi(\pi, \rho) - \frac{1}{\eta} \mathcal{B}_\phi(\pi, \pi_k), \tag{4}$$

where $\mathcal{B}_\phi$ denotes the Bregman divergence with respect to the mirror map $\phi$, which is defined as $\mathcal{B}_\phi(x; y) = \phi(x) - \phi(y) - \langle \nabla \phi(y), x - y \rangle$ with $\langle \cdot \rangle$ being the standard inner product, $\epsilon > 0$ is the regularization temperature, $\rho$ is the magnet policy (Sokota et al., 2023), and $\eta > 0$ is the stepsize (i.e., learning rate). When the mirror map $\phi$ is taken to be the negative entropy $\phi(x) = \sum_j x_j \ln x_j$, the Bregman divergence is the well-known KL divergence, and hence, we have

$$\pi_{k+1} = \arg\max_{\pi \in \Pi} \mathbb{E}_{a \sim \pi} Q_k(a) - \epsilon \mathrm{KL}(\pi, \rho) - \frac{1}{\eta} \mathrm{KL}(\pi, \pi_k). \tag{5}$$

It is easy to get that Eq. (5) possesses the closed-form solution in settings with discrete actions and unconstrained domains as follows: $\forall a \in \mathcal{A}$,

$$\pi_{k+1}^{\mathrm{KL}}(a) \propto \left[ \pi_k(a) \rho(a)^{\epsilon \eta} e^{\eta Q_k(a)} \right]^{\frac{1}{1 + \epsilon \eta}}. \tag{6}$$

We use superscript "KL" to indicate that Eq. 6 is induced with the KL divergence. The magnet policy $\rho$ is updated through $\rho_{k+1}(a) \propto \rho_k(a)^{1-\hat{\eta}} \pi_{k+1}(a)^{\hat{\eta}}$. When $\phi(x) = \frac{1}{2}\|x\|_2^2$, the Bregman divergence is the Euclidean distance. Then, we have

$$\pi_{k+1} = \arg\max_{\pi \in \Pi} \mathbb{E}_{a \sim \pi} Q_k(a) - \frac{\epsilon}{2}\|\pi - \rho\|_2^2 - \frac{1}{2\eta}\|\pi - \pi_k\|_2^2. \tag{7}$$

Similarly, we can derive the closed-form solution to Eq. (7) as follows (see Appendix B for details on the derivation): $\forall a \in \mathcal{A}$,

$$\pi_{k+1}^{\text{EU}}(a) = \frac{\epsilon\rho(a) + \frac{1}{\eta}\pi_k(a) + Q_k(a) - \frac{1}{|\mathcal{A}|}\sum_{a' \in \mathcal{A}} Q_k(a')}{(\epsilon + \frac{1}{\eta})}. \tag{8}$$

We use superscript "EU" to indicate that Eq. 8 is induced with the Euclidean distance. In addition, following (Bailey & Piliouras, 2018), we can consider the following optimization problem in each decision point:

$$\pi_{k+1} = \arg\max_{\pi \in \Pi} \eta \sum_{h=0}^{k} r_h(\pi) - \phi(\pi), \tag{9}$$

where $r_h(\pi)$ is the (expected) reward of the agent taking $\pi$. Notice that the reward is determined by the environment in single-agent settings while depends on both the environment and other agents' policies in multi-agent settings. More precisely, in multi-agent settings, $r_h(\pi) = r_h(\pi, \boldsymbol{\pi}_h^{-i})$. Then, we have another two base policy update rules, Exponential Multiplicative Weight Update ($\text{MWU}_e$, ME for short) and Linear Multiplicative Weight Update ($\text{MWU}_l$, ML for short), as follows: $\forall a \in \mathcal{A}$,

$$\pi_{k+1}^{\text{ME}}(a) = \frac{\pi_k(a)e^{\eta v_k(a)}}{\sum_{a' \in \mathcal{A}} \pi_k(a')e^{\eta v_k(a')}}, \quad \pi_{k+1}^{\text{ML}}(a) = \frac{\pi_k(a)(1 + (e^\eta - 1)v_k(a))}{\sum_{a' \in \mathcal{A}} \pi_k(a')(1 + (e^\eta - 1)v_k(a'))}, \tag{10}$$

where $v_k(a)$ denotes the reward obtained by changing the policy $\pi_k$ to a single action $a \in \mathcal{A}$.

With the above introduced four choices, we are ready to present a new policy update rule by integrating these base policies. To this end, we introduce a weight vector denoted by $\boldsymbol{\alpha} = (\alpha_1, \alpha_2, \alpha_3, \alpha_4)$ with $\sum_{j=1}^{4} \alpha_j = 1$ and $\alpha_j \geq 0$, $1 \leq j \leq 4$. Then, the new policy of the agent is computed by weighting the four base policies using $\boldsymbol{\alpha}$: $\forall a \in \mathcal{A}$,

$$\pi_{k+1}(a) = \alpha_1 \pi_{k+1}^{\text{KL}}(a) + \alpha_2 \pi_{k+1}^{\text{EU}}(a) + \alpha_3 \pi_{k+1}^{\text{ME}}(a) + \alpha_4 \pi_{k+1}^{\text{ML}}(a). \tag{11}$$

We call Eq. (11) the unified mirror descent (UMD), and the pseudo-code is shown in Algorithm 1.

The intuition behind UMD is twofold. First, although the four base policy update rules have been widely employed to solve different decision-making problems, there is no affirmative answer to the question of which one can consistently outperform all the others in terms of learning performance across all types of decision-making problems. Most of the existing theoretical results are typically limited to single-agent (Tomar et al., 2022) or two-player zero-sum games (Liu et al., 2023), and only restricted classes of games such as polymatrix and potential games have been considered while going beyond zero-sum games (Anagnostides et al., 2022). Instead of theoretically comparing these base schemes which could be difficult (if not impossible), particularly for general-sum (Song et al., 2022) and mixed cooperative-competitive settings (Xu et al., 2023), we propose a unified approach, UMD, that generalizes the base policy update rules. Intuitively, as UMD could inherit the properties of these algorithms, it could surpass or match these base methods in terms of learning performance. Second, UMD can be reduced to any of these base policy update rules by adjusting their weights. For example, when $\alpha_1 = 1$, UMD is reduced to MMD, the state-of-the-art method which unifies single-agent RL and two-player zero-sum games. In this situation, UMD could inherit the convergence guarantee of MMD in some cases such as two-player zero-sum games (Sokota et al., 2023).

## 4.2 ZERO-ORDER HYPERPARAMETER OPTIMIZATION

The key to UMD is to optimize $\boldsymbol{\alpha}$, which unfortunately, is a non-trivial task as the evaluation metric, denoted by $\mathcal{L}(\boldsymbol{\alpha})$ (the expected return or NE-Gap), is non-differentiable with respect to $\boldsymbol{\alpha}$. To address this issue, we propose two zero-order methods to optimize $\boldsymbol{\alpha}$. We adopt two representative techniques: random search follows the traditional gradient estimation algorithms (Liu et al., 2020) while GradientLess Descent (Golovin et al., 2020) uses direct search.

**Random Search (RS).** When updating the hyperparameter $\boldsymbol{\alpha}$, we first sample $M$ candidates $\{\boldsymbol{u}_i\}_i^M$ from a spherically symmetric distribution $\boldsymbol{u}_i \sim q$. Then, we compute the update as follows:

$$\boldsymbol{u}^* = -\sum\nolimits_{i=1}^{M} \mathrm{Sgn}\big[\mathcal{L}(\mathrm{Proj}(\boldsymbol{\alpha} + \mu\boldsymbol{u}_i)) - \mathcal{L}(\mathrm{Proj}(\boldsymbol{\alpha} - \mu\boldsymbol{u}_i))\big]\boldsymbol{u}_i, \tag{12}$$

where $\mathrm{Sgn}(z)$ is defined as: $\mathrm{Sgn}(z) = 1$ if $z > 0$, $\mathrm{Sgn}(z) = -1$ if $z < 0$, otherwise, $\mathrm{Sgn}(z) = 0$. $\mu$ is the smoothing parameter determining the radius of the sphere. $\mathrm{Proj}(\cdot)$ is the projection operation to ensure that $\boldsymbol{\alpha}$ is well-defined. Finally, $\boldsymbol{\alpha}$ is updated as $\boldsymbol{\alpha} \leftarrow \mathrm{Proj}(\boldsymbol{\alpha} + \boldsymbol{u}^*)$. Note that the operation $\mathrm{Sgn}(\cdot)$ plays an important role and differentiates it from vanilla RS without this operation (Wang et al., 2022). Intuitively, in the games where the performance $\mathcal{L}$ converges quickly, the magnitude of $\mathcal{L}(\mathrm{Proj}(\boldsymbol{\alpha} + \mu\boldsymbol{u}_i)) - \mathcal{L}(\mathrm{Proj}(\boldsymbol{\alpha} - \mu\boldsymbol{u}_i))$ would be too small to derive an effective update. In contrast, by using the operation $\mathrm{Sgn}(\cdot)$, the difference between the performance of $\boldsymbol{\alpha} + \mu\boldsymbol{u}_i$ and $\boldsymbol{\alpha} - \mu\boldsymbol{u}_i$ only determines the update direction, not the update magnitude.

**GradientLess Descent (GLD).** Similar to RS, when updating the hyperparameter $\boldsymbol{\alpha}$, we first sample $M$ candidates $\{\boldsymbol{u}_i\}_i^M$. However, instead of sampling from a fixed radius ($\mu$ in RS), we independently sample the candidates on spheres with various radiuses uniformly sampled from the interval $[r, R]$. Then, we follow a similar rule to compute the update as follows:

$$\boldsymbol{u}^* = -\sum\nolimits_{i=1}^{M} \mathrm{Sgn}\big[\mathcal{L}(\mathrm{Proj}(\boldsymbol{\alpha} + \boldsymbol{u}_i)) - \mathcal{L}(\boldsymbol{\alpha})\big]\boldsymbol{u}_i. \tag{13}$$

Finally, we have $\boldsymbol{\alpha} \leftarrow \mathrm{Proj}(\boldsymbol{\alpha} + \boldsymbol{u}^*)$. In contrast, in vanilla GLD (Wang et al., 2022), $\boldsymbol{\alpha}$ is updated according to the comparison between $\mathcal{L}(\boldsymbol{\alpha})$ and $\mathcal{L}(\mathrm{Proj}(\boldsymbol{\alpha} + \boldsymbol{u}_i))$: $\boldsymbol{\alpha}$ steps to the one with the best performance, or stays unchanged if none of them makes an improvement.

In addition, considering the trade-off between the learning performance and learning speed, instead of updating $\boldsymbol{\alpha}$ at each iteration, we update it every $\kappa \geq 1$ iteration (a two-timescale manner).

---

**Algorithm 1:** Unified Mirror Descent (UMD)

---

**1** Initialization: $\pi_1(a) = 1/|\mathcal{A}|, \forall a \in \mathcal{A}, \boldsymbol{\alpha} = (0.25, 0.25, 0.25, 0.25)$;
**2** **for** *iteration* $k = 1, 2, \cdots, K - 1$ **do**
**3**      Compute $\pi_{k+1}^{\mathrm{KL}}, \pi_{k+1}^{\mathrm{EU}}, \pi_{k+1}^{\mathrm{ME}}$, and $\pi_{k+1}^{\mathrm{ML}}$ through Eq. (6), (8), and (10), respectively;
**4**      **if** $k\%\kappa = 0$ **then**
**5**          Sample candidates $\{\boldsymbol{u}\}_{i=1}^M$, get $\boldsymbol{u}^*$ through RS in Eq. (12) or GLD in Eq. (13);
**6**          Update the parameters $\boldsymbol{\alpha} \leftarrow \mathrm{Proj}(\boldsymbol{\alpha} + \boldsymbol{u}^*)$;
**7**      **end**
**8** **end**
**9** **Return**: $\pi_K(a) = \alpha_1\pi_K^{\mathrm{KL}}(a) + \alpha_2\pi_K^{\mathrm{EU}}(a) + \alpha_3\pi_K^{\mathrm{ME}}(a) + \alpha_4\pi_K^{\mathrm{ML}}(a), \forall a \in \mathcal{A}$

---

## 5 EXPERIMENTS

In this section, we investigate our framework on a set of benchmark environments. We first present the experimental setups, and then the results and analysis to provide insights into our framework.

### 5.1 EXPERIMENTAL SETUPS

We consider 24 games which are divided into 5 types: single-agent, cooperative, competitive zero-sum, competitive general-sum, and mixed cooperative-competitive (MCC, for short). We construct the single-agent and MCC environments by modifying some zero-sum games. All the games are implemented in OpenSpiel (Lanctot et al., 2019). For single-agent and cooperative environments, we use the return to measure the quality of the policy/joint policy. For other cases, we use NE-Gap as the measure. In addition, to provide a clear overview of the results (Figure 1), we compute the normalized improvement of UMD versus baselines (take KL as an example): $V(\pi^{\mathrm{UMD}})/V(\pi^{\mathrm{KL}})$ for single-agent and cooperative environments, $(\mathrm{NE\text{-}Gap}(\pi^{\mathrm{Random}}) - \mathrm{NE\text{-}Gap}(\pi^{\mathrm{UMD}}))/(\mathrm{NE\text{-}Gap}(\pi^{\mathrm{Random}}) - \mathrm{NE\text{-}Gap}(\pi^{\mathrm{KL}}))$ for other environments. All methods we compare are UMD (RS), UMD (GLD), and the four base policies: KL, EU, ME, and ML. For single-agent cases, we also include Q-learning as a baseline. All experiments are performed on a machine with a 24-core Intel(R) Core(TM) i9-12900K and NVIDIA RTX A4000, and the results are obtained with 3 random seeds. The full experimental details on the games, evaluation protocol, and hyperparameters can be found in Appendix C.

## 5.2 RESULTS AND ANALYSIS

Figure 1 presents the normalized improvement of UMD (here, we refer to UMD (RS)) versus baselines (the results for UMD (GLD) can be found in Appendix D.1). Several conclusions can be drawn from the results. (i) In over 87% (21/24) games UMD performs better than or on-par with baselines, demonstrating its effectiveness in solving various types of decision-making problems. (ii) In zerosum games, UMD matches KL in all the games except Leduc. From the results, we hypothesize that UMD inherits the convergence guarantee of KL in two-player zero-sum games (Sokota et al., 2023). (iii) For some games beyond zero-sum settings, UMD can outperform the baselines. For example, in Auction, Tiny_Hanabi_B, MCC_Kuhn_A, and MCC_Kuhn_B, UMD significantly outperforms KL, which has not been observed in previous works. (iv) For the four baselines, none of them can consistently outperform all the others across different types of games, which supports the motivation of this work. For example, in Leduc, KL outperforms EU (KL > UMD > EU), while EU performs better than KL (EU > UMD > KL) in MCC_Kuhn_B.

We present the learning curves of different methods in different types of games in Figure 2 to Figure 6 (the quantitative results are given in Appendix D.1). (i) In **single-agent** cases (Figure 2), all the methods are comparable and outperform the vanilla Q-learning algorithm, showing that they can effectively solve single-agent problems. (ii) In **cooperative** settings (Figure 3), all the methods except EU and UMD (GLD) in Tiny_Hanabi_A can converge to the optimal value of the game, showing that they are effective in solving cooperative games. Surprisingly, in game B, C, and D, KL converges slower than other methods. (iii) In **competitive zero-sum** games (Figure 4), KL outperforms other methods in Kuhn and Leduc. For all the other games, UMD (RS) and KL can consistently converge to the approximate NE (low NE-Gap), while other methods can struggle or even diverge in some of the games. Typically, UMD (RS) performs better than UMD (GLD). In addition, although KL is the state-of-the-art method in (two-player) zero-sum games, it converges slower than UMD and other methods in some of the games. (iv) In **competitive general-sum** games (Figure 5), a surprising observation is that both UMD (RS) and UMD (GLD) can consistently converge to approximate NE in all the games, and in Auction, they significantly outperform KL and other methods. (v) In **mixed cooperative-competitive** cases (Figure 6), UMD (RS) can consistently converge to the approximate NE in all the games. In MCC_Kuhn_A and MCC_Kuhn_B, UMD (RS) significantly surpasses KL both in terms of convergence speed and the final NE-Gap. **In summary**, UMD (RS) can effectively solve all types of (tabular) decision-making problems, i.e., either achieving the optimal return in single-agent and cooperative cases or finding approximate NE in other cases. Moreover, in some of the games, UMD (RS)/UMD (GLD) can significantly outperform all the baselines.

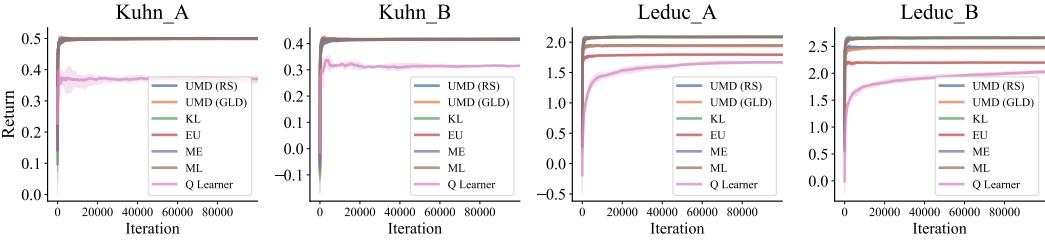

Figure 2: Experimental results for **single-agent** environments.

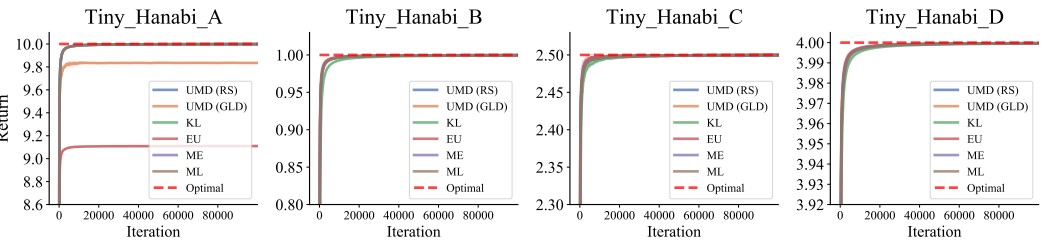

Figure 3: Experimental results for multi-agent **cooperative** environments.

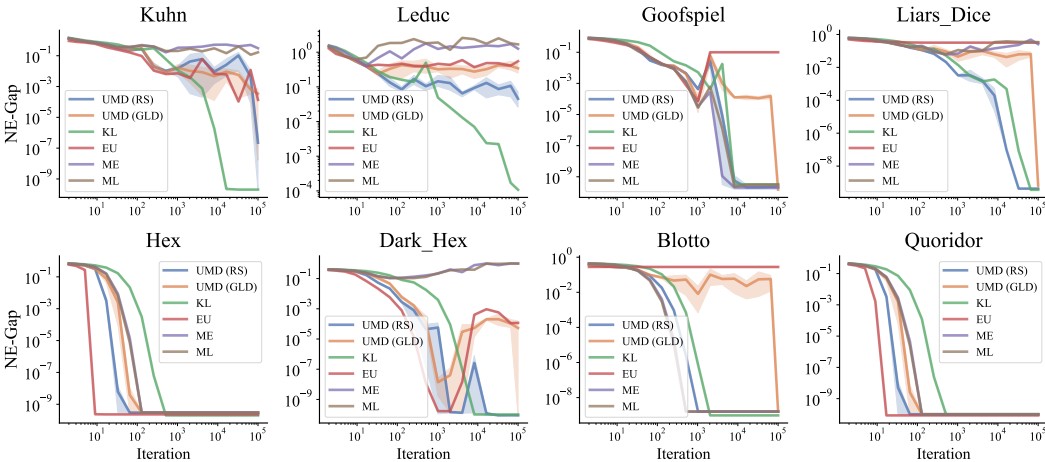

Figure 4: Experimental results for multi-agent **competitive zero-sum** environments.

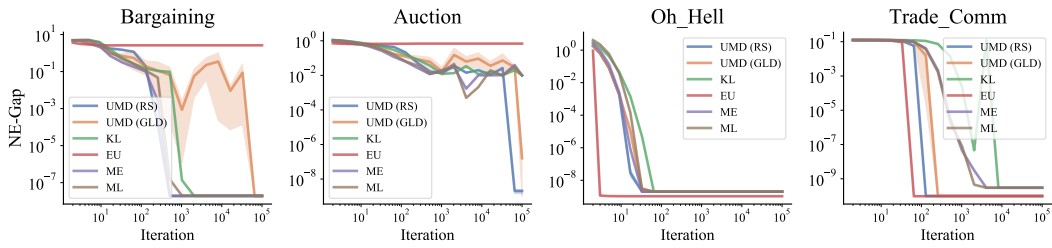

Figure 5: Experimental results for multi-agent **competitive general-sum** environments.

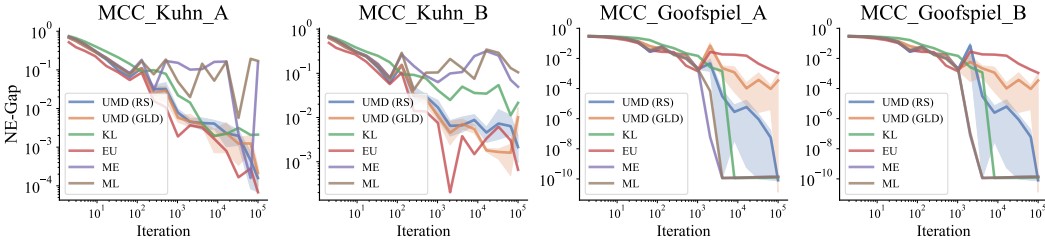

Figure 6: Experimental results for multi-agent **mixed cooperative-competitive** environments.

The key to UMD is the optimization of $\boldsymbol{\alpha}$. Intuitively, an effective HPO method should be able to identify which one of the policy update rules performs best and then assign a larger weight to this policy update rule. To verify that our proposed RS/GLD satisfies this requirement, we present the performance of different methods along with the evolution of the weights of different baselines over the learning process in Figure 7. In the left figure, we can see that when using vanilla RS/GLD (v-RS/v-GLD), UMD cannot converge to the approximate NE of the game, showing that the proposed RS/GLD is indispensable for the success of UMD. In the middle left figure, we can see that at the early stage of learning, the NE-Gap of all four base policies decreases. However, at the latter stage, EU converges to a high NE-Gap. In this situation, the weight assigned to EU should be decreased, which was exactly observed in RS and GLD in the middle right figure, demonstrating that RS and GLD can quickly adjust the weights assigned to the base policies. In the right figure, we can see that the vanilla RS and GLD cannot efficiently leverage the performance difference between the base policies to optimize the weights, leading to the failure of finding the approximate NE of the game. In addition, RS typically performs better than GLD. We hypothesize that RS is more efficient in exploring the parameter space as it uses more samples ($\boldsymbol{\alpha} + \mu\boldsymbol{u}_i$ and $\boldsymbol{\alpha} - \mu\boldsymbol{u}_i$) to get the update

direction $\boldsymbol{u}^*$ (2 times more than GLD which only involves $\boldsymbol{\alpha} + \boldsymbol{u}_i$). It is worth noting that although RS uses more samples, it does not introduce much extra computational cost compared to GLD. In Appendix D.3, we present the wall-clock time of one iteration of each method to support this claim. In fact, UMD (RS) and UMD (GLD) are still computationally efficient even compared to the four baselines. Figure 7 is obtained in Goofspiel, and more results can be found in Appendix D.2.

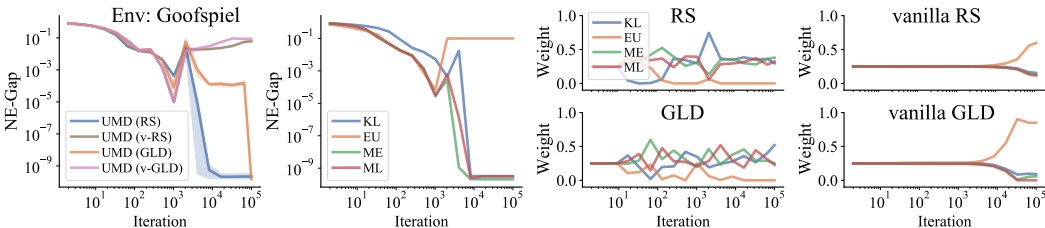

Figure 7: Comparison between RS/GLD and v-RS/v-GLD.

We also perform some ablation studies on the parameters in RS/GLD: $\kappa$, $M$, and $\mu$. Here, we only focus on $\mu$, and the results are shown in Figure 8: for single-agent and cooperative cases, $\mu$ has very little influence on the learning performance, while for other settings, different games may have different optimal $\mu$. It is worth noting that though different games may require different $\mu$, it is the only hyperparameter that requires some effort for tuning, which is also one of the advantages of our approach. For $\kappa$ and $M$, the results can be found in Appendix D.2.

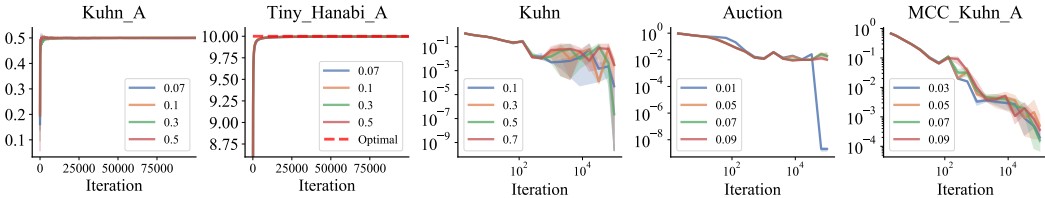

Figure 8: Influence of $\mu$ on the learning performance.

## 6 Conclusions and Future Directions

In this work, we make the first attempt to develop a single algorithm to effectively handle all types of decision-making problems under the tabular setting, including single-agent, cooperative, competitive, and mixed cooperative-competitive cases. The contributions are threefold. First, we propose a unified mirror descent (UMD) algorithm by weighting multiple base policies induced by different mirror maps to compute the new policy of an agent at each iteration. UMD is easy to extend to include new policy update rules with only minimal modifications required. Second, to optimize the weights of different base policies, we devise a simple yet effective zero-order method in which the improvement of learning performance is used to only determine the update direction of the weights rather than the update magnitude, which is more efficient than existing zero-order methods. Finally, we perform extensive experiments on 24 benchmark environments. The results show that in over 87% games UMD performs better than or on-par with baselines, demonstrating that UMD could serve as an effective unified approach for all types of (tabular) decision-making problems. Last but not least, our experiments, to our knowledge, also provide the first comprehensive empirical study of all types of (tabular) decision-making problems under a single algorithmic framework.

In this work, we focus on the decision-making problems under the tabular setting. Thus, the environments in our experiments are relatively small and simple. In future works, we may consider more complex environments where tabular representation may be a struggle (e.g., high memory and time requirements, impossible to enumerate the state space). In this situation, we need to consider a more powerful representation of the policy such as a neural network-based policy (Mnih et al., 2015), and thus, devising a single deep reinforcement learning (deep RL) algorithm to handle all types of (not restricted to tabular but more complex) decision-making problems is necessary.

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

## A    MORE DISCUSSION

In this work, we focus on the tabular setting. The reasons are twofold. First, most of the existing theoretical and algorithmic results in single-agent RL and zero-sum games were established under the tabular setting, while for more complicated general-sum and mixed cooperative-competitive cases, to our knowledge, very few results have been achieved even under the tabular setting, except for some classes of games such as polymatrix or potential games. Second, even under the tabular setting, it remains elusive whether there is a policy update rule that can consistently outperform all the others when applying different policy update rules to different decision-making problems. In view of these facts, we believe it is worthy to investigate the performance of contemporary mirror descent (MD) methods in solving various types of tabular decision-making problems before considering more complex settings where tabular representation is a struggle.

Another possible concern is that there could be other efficient algorithms for general-sum and mixed cooperative-competitive settings. In this work, we choose to adopt MD methods as they possess the last-iterate convergence property, which could be the main advantage over other possible approaches such as Double Oracle (McMahan et al., 2003) and PSRO (Lanctot et al., 2017) which show a time average convergence in some games. Indeed, to quote from (Anagnostides et al., 2022): *Last-iterate convergence is also central in economics, and goes back to the fundamental question of what it really means to "learn in a game". Indeed, it is unclear how a time average guarantee is meaningful from an economic standpoint.* In addition, it could be cumbersome and non-trivial to get an average policy when deep neural networks are employed to represent the policies, which could be inefficient in solving complex problems such as the training of GAN (Goodfellow et al., 2020). Although the last-iterate convergence may not be a universal phenomenon in games (Anagnostides et al., 2022), our empirical results show that our proposed UMD algorithm works well across different types of decision-making problems, including general-sum and mixed cooperative-competitive settings.

Last, as mentioned in Section 6, we could consider developing a single deep RL algorithm when going beyond the tabular setting. This would be possible as MD has been connected to PPO (Schulman et al., 2017) and its different variants such as MDPO (Tomar et al., 2022) and KL-PPO (Hsu et al., 2020), as given in (Sokota et al., 2023). Furthermore, independent PPO (de Witt et al., 2020; Sun et al., 2023) has been shown to be effective in single-agent RL and cooperative multi-agent RL. As a consequence, it is worth investigating whether independent PPO can be also applied to effectively solve more complex decision-making problems including multi-agent competitive and mixed cooperative-competitive settings, which we leave for future work.

## B    DERIVATION OF CLOSED-FORM SOLUTIONS

Here we present the derivation of the closed-form solution of $\pi_{k+1}^{\text{EU}}$. For the closed-form solutions of $\pi_{k+1}^{\text{KL}}$, $\pi_{k+1}^{\text{ME}}$ and $\pi_{k+1}^{\text{ML}}$, please refer to (Sokota et al., 2023; Bailey & Piliouras, 2018).

Consider the problem (7), we need to optimize the following objective:

$$\sum_{a \in \mathcal{A}} \pi(a) Q_k(a) - \frac{\epsilon}{2} \sum_{a \in \mathcal{A}} (\pi(a) - \rho(a))^2 - \frac{\beta}{2} \sum_{a \in \mathcal{A}} (\pi(a) - \pi_k(a))^2, \tag{14}$$

with the constraint $\sum_{a \in \mathcal{A}} \pi(a) = 1$, where $\beta = \frac{1}{\eta}$ for convenience. We can use Lagrange multiplier to get the following objective:

$$\sum_{a \in \mathcal{A}} \pi(a) Q_k(a) - \frac{\epsilon}{2} \|\pi - \rho\|_2^2 - \frac{\beta}{2} \|\pi - \pi_k\|_2^2 + \lambda (1 - \sum_{a \in \mathcal{A}} \pi(a)). \tag{15}$$

Taking the derivative of both $\pi$ and $\lambda$, we have:

$$Q_k(a) - \epsilon(\pi(a) - \rho(a)) - \beta(\pi(a) - \pi_k(a)) - \lambda = 0, \forall a \in \mathcal{A}, \tag{16}$$

$$\sum_{a \in \mathcal{A}} \pi(a) = 1. \tag{17}$$

Therefore from Eq. (16), we have:

$$\pi(a) = \frac{\epsilon \rho(a) + \beta \pi_k(a) + Q_k(a) - \lambda}{(\epsilon + \beta)}. \tag{18}$$

Substituting the above equation to Eq. (17), we have:

$$\sum_{a \in \mathcal{A}} \frac{\epsilon \rho(a) + \beta \pi_k(a) + Q_k(a) - \lambda}{(\epsilon + \beta)} = 1, \tag{19}$$

$$\sum_{a \in \mathcal{A}} \epsilon \rho(a) + \beta \pi_k(a) + Q_k(a) = (\epsilon + \beta) + \sum_{a \in \mathcal{A}} \lambda, \tag{20}$$

$$\lambda = \frac{\sum_{a \in \mathcal{A}} Q_k(a)}{|\mathcal{A}|}. \tag{21}$$

Note that $\sum_{a \in \mathcal{A}} \epsilon \rho(a) + \beta \pi_k(a) = \epsilon + \beta$. Then we can compute the new policy as follows:

$$\begin{aligned}
\pi(a) &= \frac{\epsilon \rho(a) + \beta \pi_k(a) + Q_k(a) - \frac{1}{|\mathcal{A}|} \sum_{a' \in \mathcal{A}} Q_k(a')}{(\epsilon + \beta)} \\
&= \frac{\epsilon \rho(a) + \frac{1}{\eta} \pi_k(a) + Q_k(a) - \frac{1}{|\mathcal{A}|} \sum_{a' \in \mathcal{A}} Q_k(a')}{(\epsilon + \frac{1}{\eta})}.
\end{aligned} \tag{22}$$

Theoretically, we note that by choosing the suitable values for $\epsilon$ and $\eta$, we can always ensure that $\pi$ is well-defined, i.e., $\pi(a) \geq 0, \forall a \in \mathcal{A}$. In experiments, we can use a projection operation to ensure this condition ($\xi = 1e - 10$ is used to avoid division by zero):

$$\pi_{k+1}^{\text{EU}}(a) = \frac{\max\{0, \pi(a)\} + \xi}{\sum_{a' \in \mathcal{A}} \max\{0, \pi(a')\} + \xi}. \tag{23}$$

## C EXPERIMENTAL DETAILS

In this section, we present the details of the games, evaluation methods, and hyperparameters.

### C.1 GAMES

Table 1 gives an overview of all the games used in our experiments. For cooperative (coop.), competitive zero-sum, and competitive general-sum (gene.-sum) settings, all the games are implemented in OpenSpiel (Lanctot et al., 2019). For single-agent and mixed cooperative-competitive (coop.-comp.) settings, we obtain the games by modifying the original games in OpenSpiel.

**Single-Agent.** Consider a two-player Kuhn poker game. To obtain a single-agent counterpart, we fix one player's policy as the uniform policy (the background player) while only updating the other player's policy (the focal player) at each iteration. In Kuhn_A, player 1 is selected as the focal player while in Kuhn_B, player 2 is chosen as the focal player. Similarly, we can get Leduc_A and Leduc_B in the same manner.

**Cooperative.** For cooperative environments, we focus on two-player tiny Hanabi games (Foerster et al., 2019; Sokota et al., 2021). A suite of six games is available at `https://github.com/ssokota/tiny-hanabi`. We chose four of the six games as our testing environments and rename them to A through D. The payoff matrices along with the optimal values of these games are given in Figure 9. For implementation, these games are easy to obtain by setting the three parameters: `num_chance`, `num_actions`, and `payoff`, in OpenSpiel. For `num_chance`, they are 2, 2, 2, and 1, respectively. For `num_actions`, they are 3, 2, 2, and 2, respectively.

**Competitive Zero-Sum and General-Sum.** All the zero-sum and general-sum games have been implemented in OpenSpiel. The configurations of different games are given in the second column in Table 1. Note that in contrast to most of the existing works which only focus on two-player (zero-sum) settings, we set the number of players to more than two players in some of the games: Kuhn, Goofspiel, and Oh_Hell are three-player games.

**Mixed Cooperative-Competitive (MCC).** Consider a three-player Kuhn poker game. To obtain an MCC counterpart, we partition the three players into two teams: Team 1 includes two players while Team 2 only consists of one player (i.e., two *vs.* one). When computing the rewards of the players, in Team 1, each player will get the average reward of the team. Precisely, let $r^{\text{team}} = r^1 + r^2$ denote the team reward which is the sum of the original rewards of the two team members. Then, the true

Table 1: The benchmark games used in experiments.

| Type | Name of Game w/ Configuration | Shorthand |
|---|---|---|
| Single-Agent | single_agent_kuhn_2p_game_a | Kuhn_A |
| | single_agent_kuhn_2p_game_b | Kuhn_B |
| | single_agent_leduc_2p_game_a | Leduc_A |
| | single_agent_leduc_2p_game_b | Leduc_B |
| Coop. | tiny_hanabi_game_a | Tiny_Hanabi_A |
| | tiny_hanabi_game_b | Tiny_Hanabi_B |
| | tiny_hanabi_game_c | Tiny_Hanabi_C |
| | tiny_hanabi_game_d | Tiny_Hanabi_D |
| Zero-Sum | kuhn_poker(players=3) | Kuhn |
| | leduc_poker(players=2) | Leduc |
| | goofspiel(players=3) | Goofspiel |
| | liars_dice(dice_sides=4) | Liars_Dice |
| | hex(board_size=2) | Hex |
| | dark_hex(board_size=2,gameversion=adh) | Dark_Hex |
| | blotto(players=2,coins=4,fields=4) | Blotto |
| | quoridor(players=2,board_size=2) | Quoridor |
| Gene.-Sum | bargaining(max_turns=2) | Bargaining |
| | first_sealed_auction | Auction |
| | oh_hell(players=3,num_suits=2,num_cards_per_suit=2) | Oh_Hell |
| | trade_comm(num_items=2) | Trade_Comm |
| Mixed Coop.-Comp. | mix_kuhn_3p_game_a | MCC_Kuhn_A |
| | mix_kuhn_3p_game_b | MCC_Kuhn_B |
| | mix_goofspiel_3p_game_a | MCC_Goofspiel_A |
| | mix_goofspiel_3p_game_b | MCC_Goofspiel_B |

rewards of the two players are $\tilde{r}^1 = \tilde{r}^2 = r^{\text{team}}/2$. In MCC_Kuhn_A, Team 1 includes players 1 and 2 (i.e., $\{1, 2\}$ *vs.* 3), while in MCC_Kuhn_B, Team 1 includes players 1 and 3 (i.e., $\{1, 3\}$ *vs.* 2). Similarly, we can get MCC_Goofspiel_A and MCC_Goofspiel_B in the same manner.

## C.2 EVALUATION PROTOCOL

For **Single-Agent** cases, evaluating the performance of the focal player's policy is easy as we only need to estimate the expected return obtained by the focal player's policy while regarding the other agent as a part of the environment (i.e., from a single agent's perspective). For **Cooperative** settings, we can easily obtain the team return. For **Competitive Zero-Sum and General-Sum** settings, we can easily compute the NE-Gap by using the built-in implementation in OpenSpiel.

For **Mixed Cooperative-Competitive (MCC)** settings, to compute the NE-Gap, we need to compute the best response policy of the team, i.e., a joint policy of the team members, rather than the policy of a single agent. This is incompatible with the built-in implementation in OpenSpiel, which only computes the best response policy of a single agent. In other words, if we directly adopt the built-in implementation, the NE-Gap will correspond to the original three-player game, not the modified game. Unfortunately, computing the exact joint policy of the team members is not easy in practice. Nevertheless, it is worth noting that from our experiments, we found that KL (and other base policy update rules) can effectively solve cooperative decision-making problems. As a result,

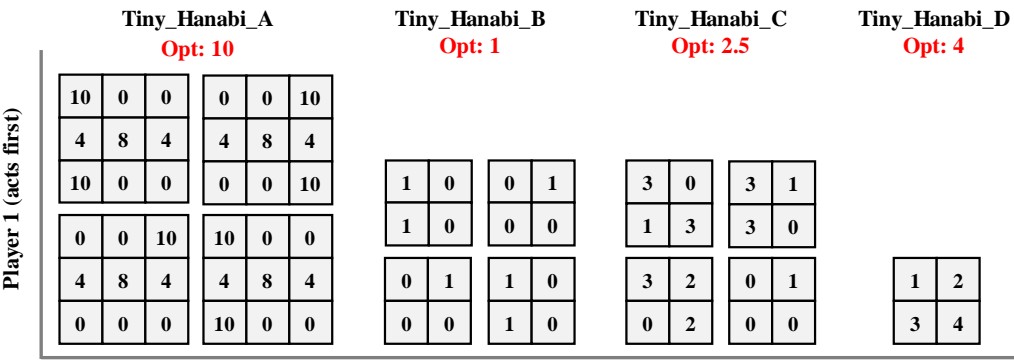

Figure 9: Payoff matrices and optimal values of the four (cooperative) tiny Hanabi games.

we can apply KL (or other base policy update rules) to compute the approximate best response of the team since it is a purely cooperative environment from the team's perspective (the other team's policy is fixed when computing the best response of the team). For a team that only has a single player, we use the built-in implementation in OpenSpiel to compute the exact best response policy of the player. In summary, during the policy learning process, when evaluation of the current joint policy is needed, we use KL as a subroutine to compute a team's approximate best response while using built-in implementation to compute a single player's exact best response. In the KL subroutine, the starting point of the best response is set to the current joint policy of the team members. In experiments, to balance the accuracy of the best response and running time, the number of updates in the KL subroutine is set to 100 (the returned joint policy can be also called a better response).

## C.3 HYPERPARAMETERS

Table 2 provides the default values of hyperparameters used in different games. In RS, the spherically symmetric distribution $q$ is a standard multivariate normal distribution $\mathbb{N}(\mathbf{0}, \mathbf{I})$. The $\text{Proj}(\boldsymbol{\alpha})$ is defined as $\alpha_i = \frac{\alpha_i + \xi}{\sum_{j=1}^4 \alpha_j + \xi}, 1 \le i \le 4$, where $\xi = 1e - 10$. In addition, in our experiments, we perform ablation studies on $\mu$, $M$, and $\kappa$, while all other parameters are kept constant.

Table 2: Hyperparameters.

| Game | $K$ | $\epsilon$ | $\eta$ | $\hat{\eta}$ | $\mu$ | $r$ | $R$ | $M$ | $\kappa$ |
|---|---|---|---|---|---|---|---|---|---|
| Kuhn_A | 100000 | 1 | 0.1 | 0.05 | 0.1 | 0.1 | 2 | 5 | 10 |
| Kuhn_B | 100000 | 1 | 0.1 | 0.05 | 0.1 | 0.1 | 2 | 5 | 10 |
| Leduc_A | 100000 | 1 | 0.1 | 0.05 | 0.1 | 0.1 | 2 | 5 | 10 |
| Leduc_B | 100000 | 1 | 0.1 | 0.05 | 0.1 | 0.1 | 2 | 5 | 10 |
| Tiny_Hanabi_A | 100000 | 1 | 0.1 | 0.05 | 0.1 | 0.1 | 2 | 5 | 10 |
| Tiny_Hanabi_B | 100000 | 1 | 0.1 | 0.05 | 0.1 | 0.1 | 2 | 5 | 10 |
| Tiny_Hanabi_C | 100000 | 1 | 0.1 | 0.05 | 0.1 | 0.1 | 2 | 5 | 10 |
| Tiny_Hanabi_D | 100000 | 1 | 0.1 | 0.05 | 0.1 | 0.1 | 2 | 5 | 10 |
| Kuhn | 100000 | 1 | 0.1 | 0.05 | 0.5 | 0.1 | 2 | 5 | 10 |
| Leduc | 100000 | 1 | 0.1 | 0.05 | 0.03 | 0.1 | 2 | 5 | 10 |
| Goofspiel | 100000 | 1 | 0.1 | 0.05 | 0.1 | 0.1 | 2 | 5 | 10 |
| Liars_Dice | 100000 | 1 | 0.1 | 0.05 | 0.05 | 0.1 | 2 | 5 | 10 |
| Hex | 100000 | 1 | 0.1 | 0.05 | 0.1 | 0.1 | 2 | 5 | 10 |
| Dark_Hex | 100000 | 1 | 0.1 | 0.05 | 0.1 | 0.1 | 2 | 5 | 10 |
| Blotto | 100000 | 1 | 0.1 | 0.05 | 0.1 | 0.1 | 2 | 5 | 10 |
| Quoridor | 100000 | 1 | 0.1 | 0.05 | 0.1 | 0.1 | 2 | 5 | 10 |
| Bargaining | 100000 | 1 | 0.1 | 0.05 | 0.01 | 0.1 | 2 | 5 | 10 |
| Auction | 100000 | 1 | 0.1 | 0.05 | 0.01 | 0.1 | 2 | 5 | 10 |
| Oh_Hell | 100000 | 1 | 0.1 | 0.05 | 0.3 | 0.1 | 2 | 5 | 10 |
| Trade_Comm | 100000 | 1 | 0.1 | 0.05 | 0.3 | 0.1 | 2 | 5 | 10 |
| MCC_Kuhn_A | 100000 | 1 | 0.1 | 0.05 | 0.07 | 0.1 | 2 | 5 | 10 |
| MCC_Kuhn_B | 100000 | 1 | 0.1 | 0.05 | 0.05 | 0.1 | 2 | 5 | 10 |
| MCC_Goofspiel_A | 100000 | 1 | 0.1 | 0.05 | 0.5 | 0.1 | 2 | 5 | 10 |
| MCC_Goofspiel_B | 100000 | 1 | 0.1 | 0.05 | 0.5 | 0.1 | 2 | 5 | 10 |

# D   MORE EXPERIMENTAL RESULTS

In this section, we provide more experimental results and analysis to support the conclusions of this work and deepen the understanding of our approach. For ablation studies, we perform experiments on some of the games (not all) as other games may require a relatively long running time (e.g., the mixed cooperative-competitive environments), as shown in Appendix D.3.

## D.1   LEARNING PERFORMANCE

**Normalized Improvement.** In Figure 1, we show the normalized improvement of UMD (RS) versus baselines. Here, we provide some explanations of the magnitude of the quantitative values. Recall that the normalized improvement is computed as (take KL as an example) (NE-Gap($\pi^{\text{Random}}$) − NE-Gap($\pi^{\text{UMD}}$))/(NE-Gap($\pi^{\text{Random}}$) − NE-Gap($\pi^{\text{KL}}$)) ($\pi^{\text{Random}}$ is the random initialized policy). In some environments, NE-Gap($\pi^{\text{UMD}}$) and NE-Gap($\pi^{\text{KL}}$) could be very small, e.g., $< 1e - 8$. As a result, the normalized improvement would only slightly deviate from 1 as the NE-Gap($\pi^{\text{Random}}$) is typically relatively large, e.g., $> 1$. However, it is worth highlighting that this does not mean that the improvement is not significant. For example, in Auction, as shown in Figure 5 (the learning curves), the final NE-Gap for UMD (RS) and KL are respectively 2.0340e-9 and 1.0000e-2, which shows that UMD (RS) significantly outperforms KL in an order of 7 magnitude, even though the value of the normalized improvement is 1.0081 (NE-Gap($\pi^{\text{Random}}$)=1.2493).

In Figure 10, we present the normalized improvement of UMD (GLD) versus baselines. Compare Figure 1 and 10, we can see that UMD (RS) typically performs better than UMD (GLD), demonstrating that RS is more efficient than GLD in optimizing the weights of different base policies. For example, in Tiny_Hanabi_A, MCC_Goofspiel_A, and MCC_Goofspiel_B, UMD (GLD) apparently lags behind KL as shown in Figure 10, which is not the case shown in Figure 1 where UMD (RS) matches KL. In Leduc, the normalized improvement of UMD (GLD) versus KL is 0.8557, while it is 0.9809 for UMD (RS) versus KL. The quantitative values of the normalized improvement of UMD (RS) (Figure 1) and UMD (GLD) (Figure 10) versus the baselines are provided in Table 3.

Note that for Dark_Hex, from Figure 4 (the learning curves), we can see that ME and ML diverge as the learning process progresses. As a consequence, the final NE-Gap of ME/ML is even larger than a random policy. In this case, the normalized improvement will be negative as NE-Gap($\pi^{\text{Random}}$) − NE-Gap($\pi^{\text{UMD}}$) $> 0$ while NE-Gap($\pi^{\text{Random}}$) − NE-Gap($\pi^{\text{ME}}$) $< 0$, which is inconvenient to draw the bar for this game in Figure 1 and Figure 10. Without loss of generality, when computing the normalized improvement of UMD (RS) and UMD (GLD) versus ME and ML for Dark_Hex, we simply use a number greater than 1 (i.e., 1.10(*) in Table 3). Notice that this number does not stand for the true normalized improvement, but is only used to indicate that UMD (RS) and UMD (GLD) definitely outperform ME and ML as ME and ML diverge in this game.

In Table 4, we present the quantitative values of the converged performance of different methods in different types of environments (Figure 2 to Figure 6). Numbers in bold indicate performance ranking in the top three. From the results, we can see that UMD (RS) can effectively solve different types of games, and in some games, it can significantly outperform the baselines.

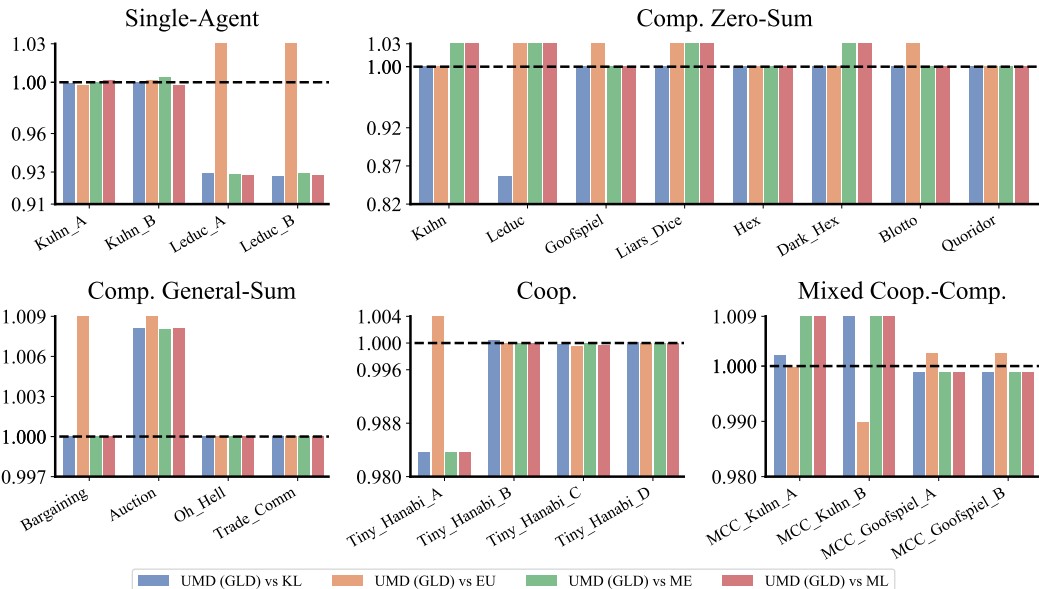

Figure 10: The Y-axis is the normalized improvement of UMD (GLD) versus baselines: $> 1$ means UMD (GLD) outperforms the baselines, $= 1$ means UMD(GLD) matches the baselines, and $< 1$ means UMD (GLD) lags behind the baselines. The numbers of games in which UMD (GLD) outperforms or matches the baselines are 20 for KL, 23 for EU, 19 for ME, and 18 for ML. On average, in over 83% games, UMD (GLD) outperforms or matches the baselines. In addition, compared to Figure 1, we found that UMD (GLD) typically lags behind UMD (RS) which performs better than or on-par with the baselines in over 87% games.

Table 3: The quantitative values of the normalized improvement of UMD (RS) and UMD (GLD) versus different baselines in different games, which correspond to Figure 1 and Figure 10. $> 1$ means UMD (RS)/UMD (GLD) outperforms the baselines, $= 1$ means UMD (RS)/UMD (GLD) matches the baselines, and $< 1$ means UMD (RS)/UMD (GLD) lags behind the baselines.

| Game | UMD (RS) vs. | | | | UMD (GLD) vs. | | | |
|---|---|---|---|---|---|---|---|---|
| | KL | EU | ME | ML | KL | EU | ME | ML |
| Kuhn_A | 1.0002 | 0.9994 | 1.0010 | 1.0026 | 0.9988 | 0.9980 | 0.9996 | 1.0012 |
| Kuhn_B | 1.0032 | 1.0052 | 1.0073 | 1.0014 | 0.9996 | 1.0016 | 1.0037 | 0.9979 |
| Leduc_A | 0.9333 | 1.0846 | 0.9326 | 0.9318 | 0.9293 | 1.0800 | 0.9286 | 0.9278 |
| Leduc_B | 0.9335 | 1.1309 | 0.9359 | 0.9348 | 0.9264 | 1.1223 | 0.9287 | 0.9277 |
| Tiny_Hanabi_A | 1.0000 | 1.0976 | 1.0000 | 1.0000 | 0.9836 | 1.0796 | 0.9836 | 0.9836 |
| Tiny_Hanabi_B | 1.0004 | 0.9999 | 1.0000 | 1.0000 | 1.0004 | 0.9999 | 1.0000 | 1.0000 |
| Tiny_Hanabi_C | 1.0001 | 0.9998 | 1.0002 | 0.9999 | 0.9999 | 0.9996 | 0.9999 | 0.9996 |
| Tiny_Hanabi_D | 1.0000 | 1.0000 | 1.0000 | 1.0000 | 1.0000 | 1.0000 | 1.0000 | 1.0000 |
| Kuhn | 1.0000 | 1.0001 | 1.1753 | 1.0878 | 0.9998 | 0.9999 | 1.1751 | 1.0876 |
| Leduc | 0.9809 | 1.2775 | 2.0907 | 3.3555 | 0.8557 | 1.1143 | 1.8237 | 2.9269 |
| Goofspiel | 1.0000 | 1.1224 | 1.0000 | 1.0000 | 1.0000 | 1.1224 | 1.0000 | 1.0000 |
| Liars_Dice | 1.0000 | 1.9443 | 1.6425 | 2.0867 | 1.0000 | 1.9443 | 1.6425 | 2.0867 |
| Hex | 1.0000 | 1.0000 | 1.0000 | 1.0000 | 1.0000 | 1.0000 | 1.0000 | 1.0000 |
| Dark_Hex | 1.0000 | 1.0003 | 1.10(*) | 1.10(*) | 0.9999 | 1.0002 | 1.10(*) | 1.10(*) |
| Blotto | 1.0000 | 2.5510 | 1.0000 | 1.0000 | 1.0000 | 2.5510 | 1.0000 | 1.0000 |
| Quoridor | 1.0000 | 1.0000 | 1.0000 | 1.0000 | 1.0000 | 1.0000 | 1.0000 | 1.0000 |
| Bargaining | 1.0000 | 2.0991 | 1.0000 | 1.0000 | 1.0000 | 2.0991 | 1.0000 | 1.0000 |
| Auction | 1.0081 | 2.2287 | 1.0080 | 1.0081 | 1.0081 | 2.2287 | 1.0080 | 1.0081 |
| Oh_Hell | 1.0000 | 1.0000 | 1.0000 | 1.0000 | 1.0000 | 1.0000 | 1.0000 | 1.0000 |
| Trade_Comm | 1.0000 | 1.0000 | 1.0000 | 1.0000 | 1.0000 | 1.0000 | 1.0000 | 1.0000 |
| MCC_Kuhn_A | 1.0020 | 0.9999 | 1.1858 | 1.2091 | 1.0019 | 0.9999 | 1.1858 | 1.2090 |
| MCC_Kuhn_B | 1.0215 | 0.9984 | 1.0544 | 1.1266 | 1.0127 | 0.9898 | 1.0453 | 1.1169 |
| MCC_Goofspiel_A | 1.0000 | 1.0035 | 1.0000 | 1.0000 | 0.9989 | 1.0024 | 0.9989 | 0.9989 |
| MCC_Goofspiel_B | 1.0000 | 1.0035 | 1.0000 | 1.0000 | 0.9989 | 1.0024 | 0.9989 | 0.9989 |

Table 4: The quantitative values of performance of different methods, which corresponds to Figure 2 to Figure 6. For single-agent and cooperative cases, the larger the better, while it is the opposite for other cases. Numbers in bold indicate performance ranking in the top three. The results clearly show that UMD (RS) can effectively solve various types of (tabular) decision-making problems.

| Game | KL | EU | ME | ML | UMD (GLD) | UMD (RS) |
|---|---|---|---|---|---|---|
| Kuhn_A | **0.4998** | **0.5002** | 0.4994 | 0.4986 | 0.4992 | **0.4999** |
| Kuhn_B | **0.4165** | 0.4156 | 0.4148 | **0.4172** | 0.4163 | **0.4178** |
| Leduc_A | **2.0855** | 1.7945 | **2.0869** | **2.0889** | 1.9504 | 1.9495 |
| Leduc_B | **2.6630** | 2.1981 | **2.6563** | **2.6593** | 2.5008 | 2.4274 |
| Tiny_Hanabi_A | **9.9987** | 9.1098 | **9.9988** | **9.9987** | 9.8346 | **9.9987** |
| Tiny_Hanabi_B | **0.9993** | **0.9997** | **0.9997** | **0.9997** | 0.9996 | **0.9997** |
| Tiny_Hanabi_C | **2.4995** | **2.5000** | 2.4993 | **2.5000** | 2.4991 | **2.4998** |
| Tiny_Hanabi_D | **3.9995** | **3.9997** | **3.9997** | **3.9997** | **3.9997** | **3.9997** |
| Kuhn | **2.034e-10** | **1.358e-04** | 3.077e-01 | 1.665e-01 | 3.465e-04 | **2.261e-07** |
| Leduc | **1.073e-04** | 5.510e-01 | 1.260e-00 | 1.680e-00 | **3.427e-01** | **4.540e-02** |
| Goofspiel | 3.246e-10 | 9.990e-02 | **2.068e-10** | 3.091e-10 | **1.501e-10** | **2.146e-10** |
| Liars_Dice | **3.631e-10** | 3.181e-01 | 2.562e-01 | 3.411e-01 | **6.605e-10** | **4.283e-10** |
| Hex | **2.000e-10** | **2.312e-10** | 3.000e-10 | 3.000e-10 | 3.000e-10 | **2.999e-10** |
| Dark_Hex | **1.007e-10** | 1.162e-04 | 9.809e-01 | 9.998e-01 | **5.320e-05** | **8.766e-11** |
| Blotto | **9.666e-10** | 2.779e-01 | **1.599e-09** | **1.599e-09** | **1.599e-09** | **1.599e-09** |
| Quoridor | 1.000e-10 | **8.461e-11** | 1.000e-10 | 1.000e-10 | **9.999e-11** | **9.999e-11** |
| Bargaining | **1.927e-08** | 2.618e-00 | 1.991e-08 | 1.979e-08 | **1.845e-08** | **1.854e-08** |
| Auction | 1.000e-02 | 6.887e-01 | **9.900e-03** | 1.000e-02 | **1.596e-07** | **2.034e-09** |
| Oh_Hell | **2.000e-09** | **1.047e-09** | **2.000e-09** | **2.000e-09** | **2.000e-09** | **2.000e-09** |
| Trade_Comm | 3.000e-10 | **9.565e-11** | 3.000e-10 | 3.000e-10 | **9.999e-11** | **9.999e-11** |
| MCC_Kuhn_A | 2.100e-03 | **6.794e-05** | 1.542e-01 | 1.701e-01 | **2.129e-04** | **1.591e-04** |
| MCC_Kuhn_B | 2.140e-02 | **6.562e-04** | 4.950e-02 | 1.053e-01 | **1.010e-02** | **2.100e-03** |
| MCC_Goofspiel_A | **1.261e-10** | 1.100e-03 | 1.375e-10 | **1.374e-10** | 3.423e-04 | **8.131e-11** |
| MCC_Goofspiel_B | **1.237e-10** | 1.100e-03 | **1.378e-10** | 1.379e-10 | 3.424e-04 | **8.033e-11** |

## D.2 MORE ABLATION STUDIES

In this section, we present more ablation results to deepen the understanding of our approach.

### D.2.1 EFFECTIVENESS OF RS/GLD

As mentioned in the main text, RS/GLD can quickly adjust the weights assigned to the base policies, which is the key to the success of UMG (RS)/UMD (GLD) in solving various types of decision-making problems. In Figure 11 to Figure 15, we present, in different types of games, the performance of the four baselines (the first row) and the evolution of the weights (the second row for RS and the last row for GLD). In addition, in Figure 7 in the main text, we also present the weights obtained by the vanilla RS (v-RS) and vanilla GLD (v-GLD) in Goofspiel, which demonstrates that RS and GLD are superior over their vanilla counterparts. In Figure 16, we present more results in other games, which provides more evidence to support the conclusion.

In **single-agent** (Figure 11) and **cooperative** (Figure 12) cases, in most of the environments, since the four baselines are comparable and can achieve the optimal return, their weights do not change too much (around 1/4) over the learning process. In **competitive zero-sum** (Figure 13), **competitive general-sum** (Figure 14), and **mixed cooperative-competitive** (Figure 15) cases, RS/GLD can quickly adjust the weights of the baselines according to their performance over the learning process.

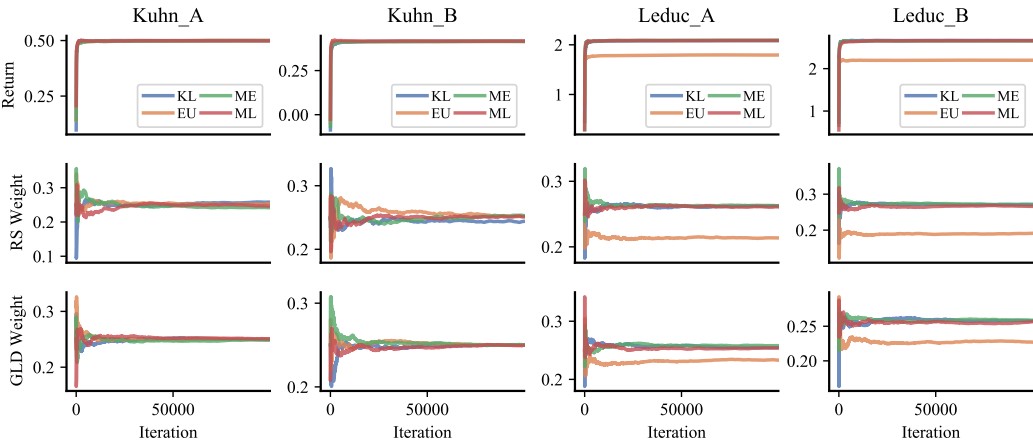

Figure 11: Weights of base policies for **single-agent** environments.

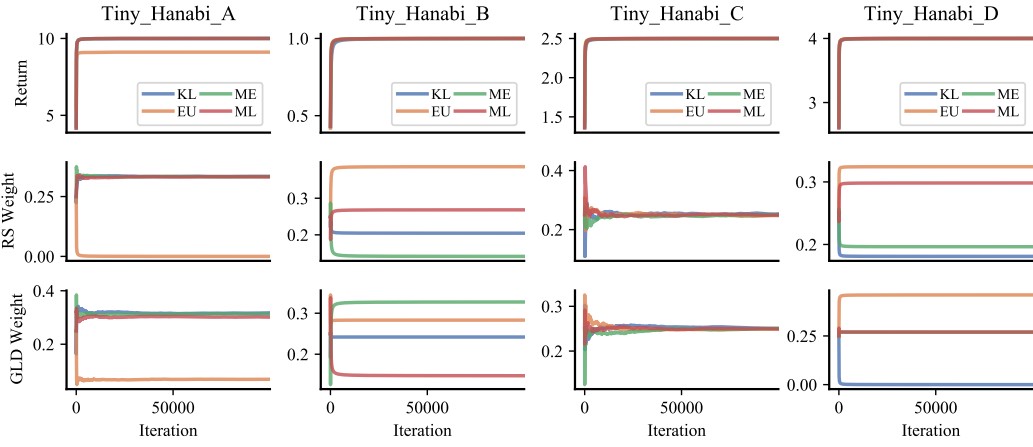

Figure 12: Weights of base policies for multi-agent **cooperative** environments.

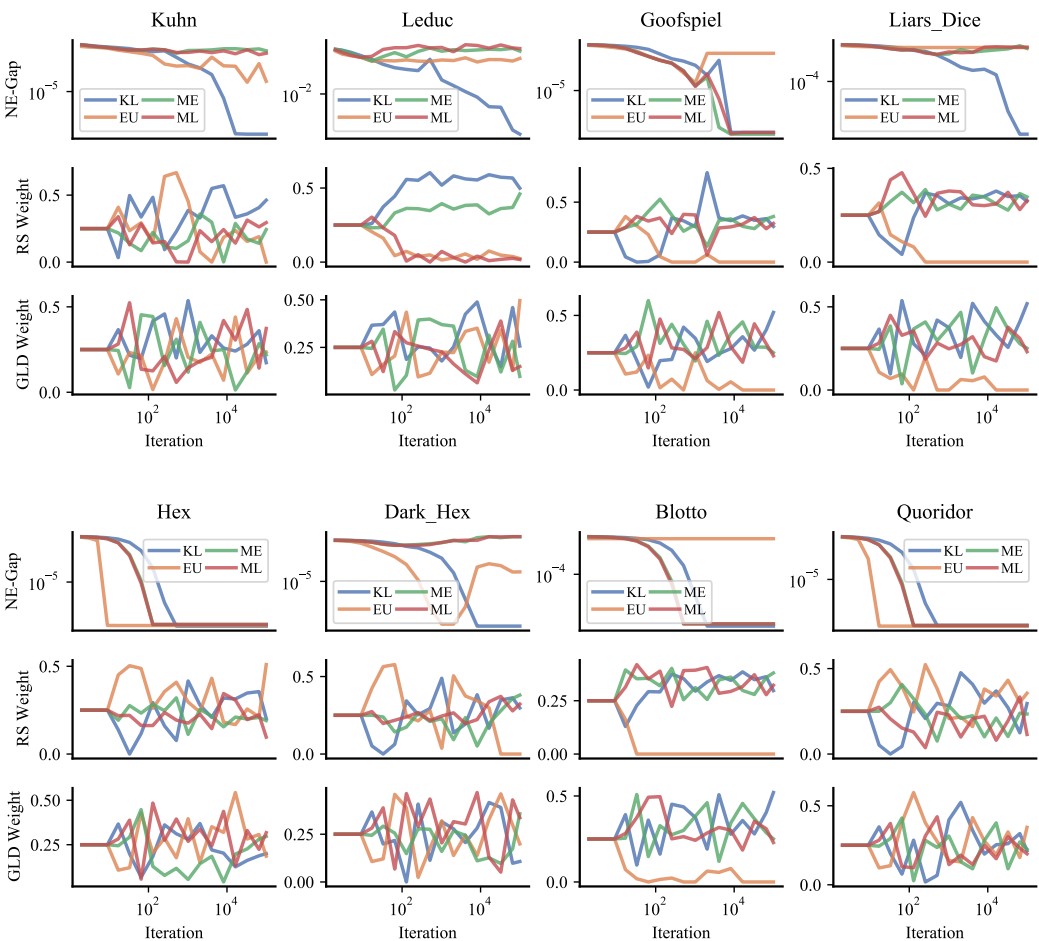

Figure 13: Weights of base policies for multi-agent **competitive zero-sum** environments.

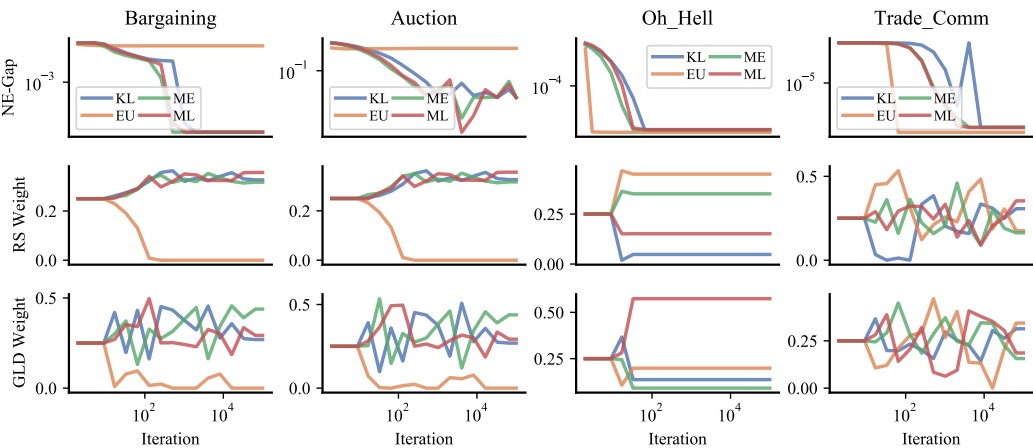

Figure 14: Weights of base policies for multi-agent **competitive general-sum** environments.

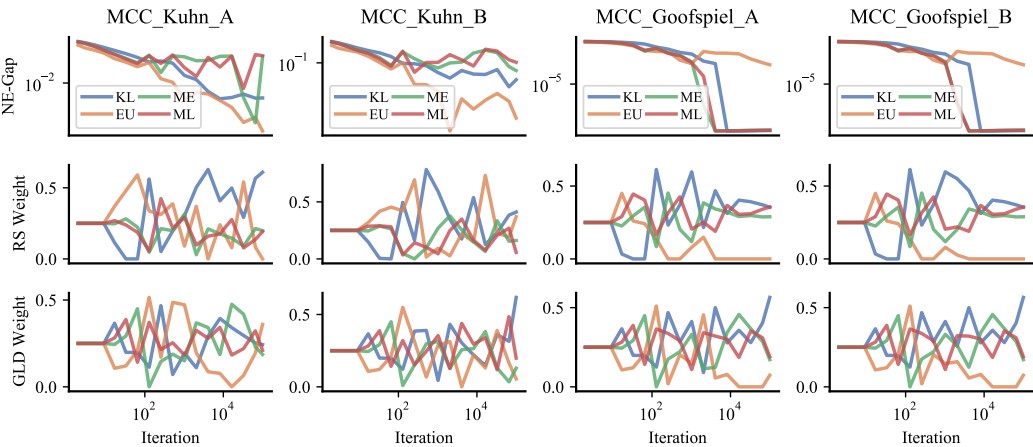

Figure 15: Weights of base policies for multi-agent **mixed cooperative-competitive** environments.

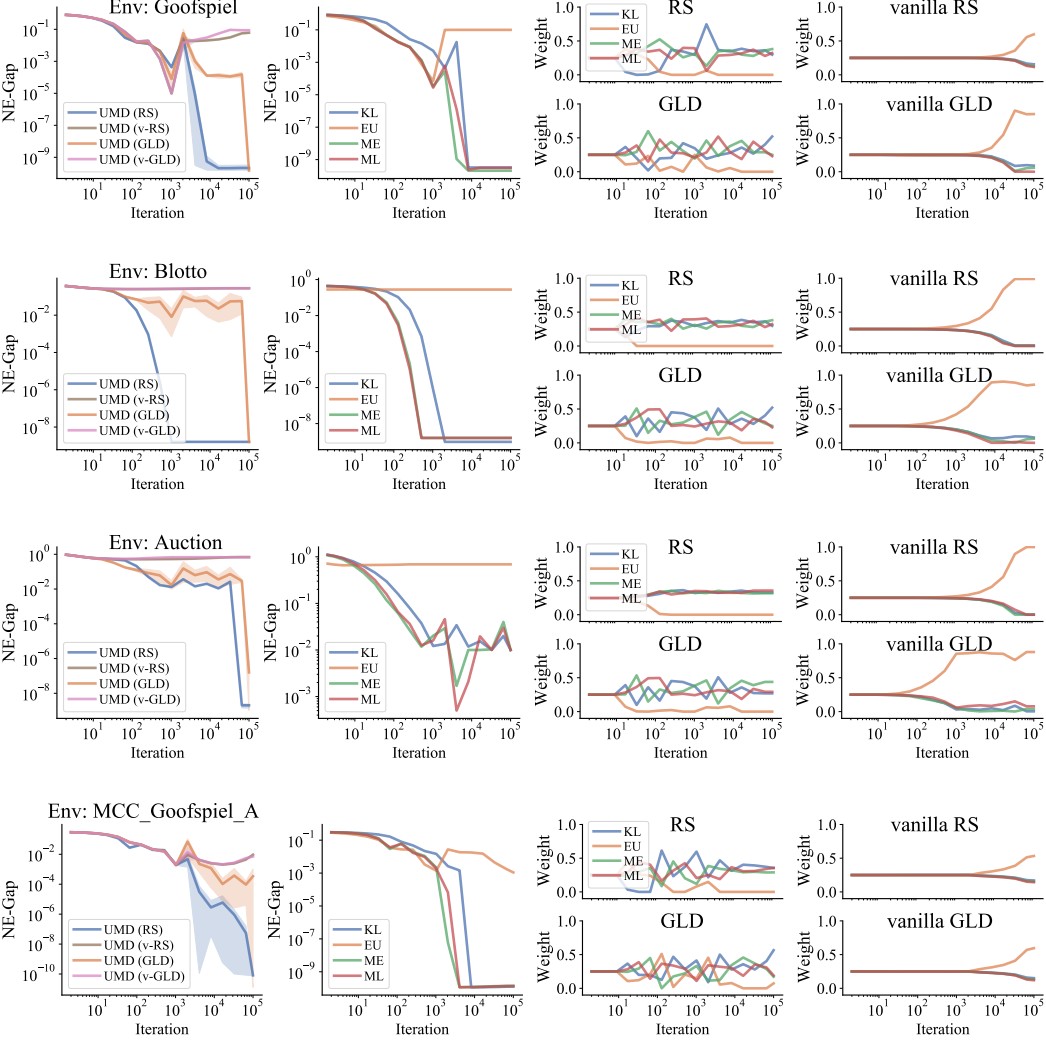

Figure 16: Comparison between RS/GLD and v-RS/v-GLD.

### D.2.2 INFLUENCE OF PARAMETERS IN RS

In Figure 8 in the main text, we present the influence of $\mu$ on the learning performance. As $\mu$ has very little influence in single-agent and cooperative cases, here, we provide more results in other cases in Figure 17. The results again verify the fact that different games may require different optimal $\mu$. However, as mentioned in the main text, this may be the only parameter that requires some effort for tuning in different games, which is one of the advantages of our approach.

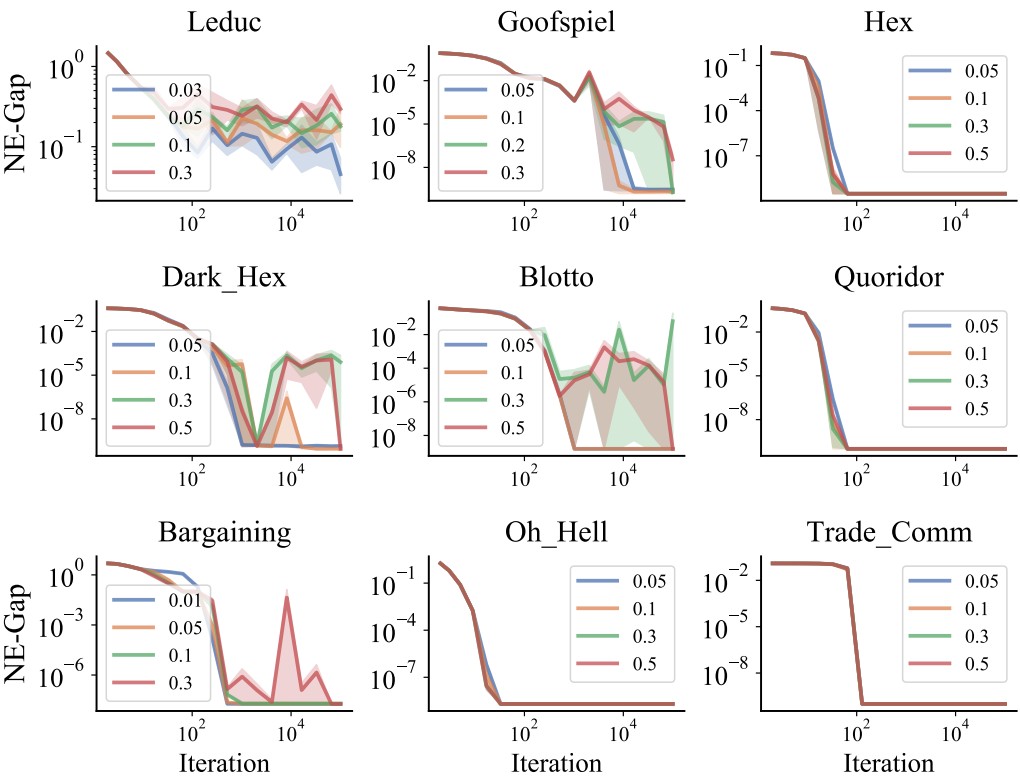

Figure 17: The influence of $\mu$ on the learning performance.

In Figure 18, we investigate the influence of the update interval $\kappa$ on the learning performance. For single-agent and cooperative cases, $\kappa$ has almost negligible influence on the learning performance of UMD (RS). For other cases, $\kappa$ may slightly influence the convergence speed but has little influence on the final performance. As a result, in our experiments, we choose $\kappa = 10$ (a two-timescale manner) as the default value to reduce the running time of the learning process (Table 5 in Appendix D.3) while not incurring much loss on the learning performance.

In Figure 19, we investigate the influence of the number of samples $M$ on the learning performance. The results show a similar phenomena with $\kappa$. Therefore, to balance the learning speed and performance, we choose a moderate number $M = 5$ as the default value.

Furthermore, we want to remark that we do not conduct thorough parameter sweeping in this work. Consequently, the default values given in Table 2 may be not always optimal. Nevertheless, from the experimental results, we can see that these default values work well across all types of (tabular) decision-making problems. This may be partly due to the fact that we are focusing on the tabular cases. For more complex settings where tabular representation is a struggle, more advancing learning techniques (e.g., deep RL) are required, which we leave for future work.

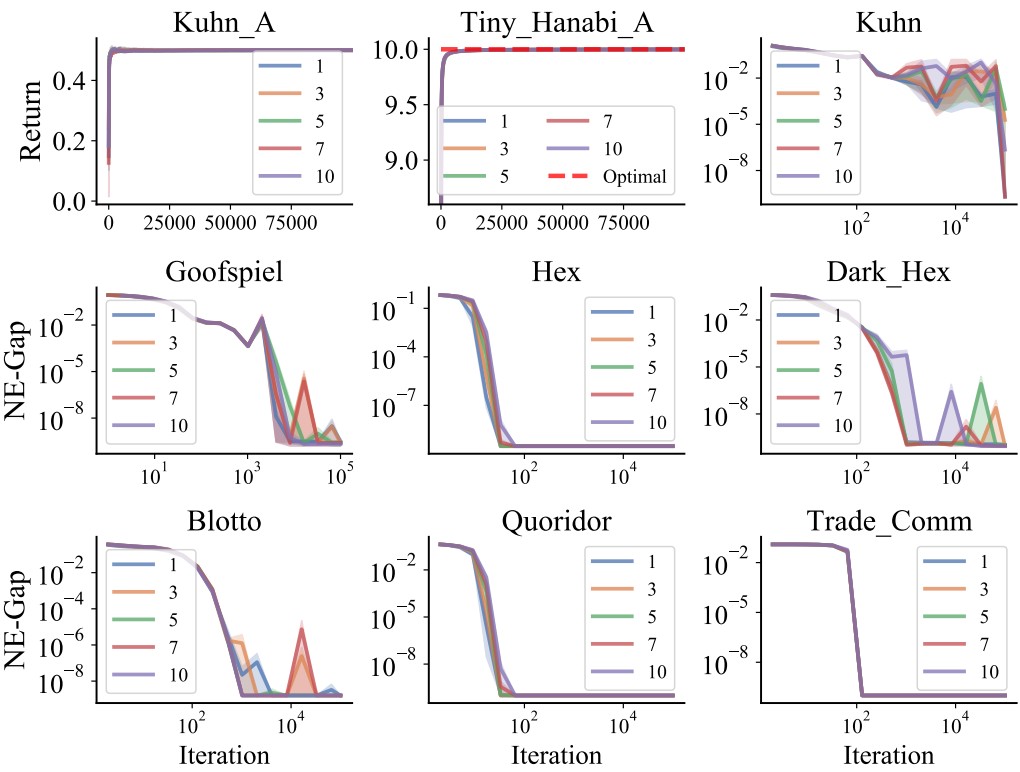

Figure 18: The influence of update interval $\kappa$ on the learning performance.

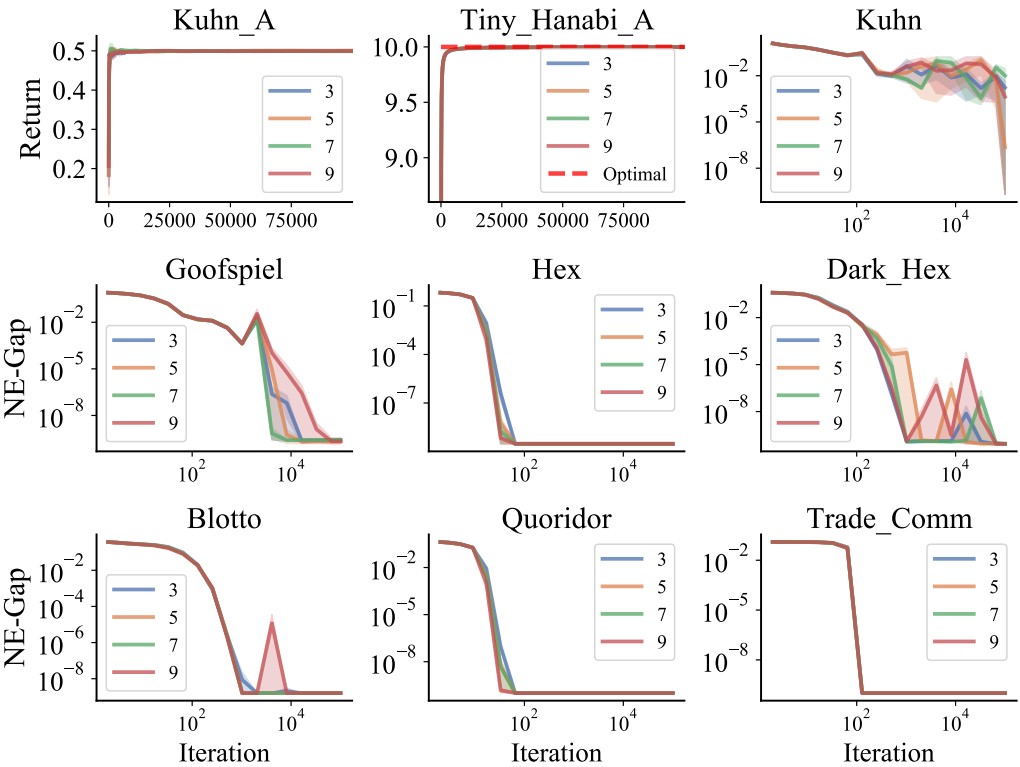

Figure 19: The influence of the number of samples $M$ on the learning performance.

## D.3 RUNNING TIME

In Table 5, we present the running time of one iteration of different methods in different types of games under the default values of hyperparameters given in Table 2. Compared to the four baselines, we can see that UMD (RS) and UMD (GLD) do not introduce much extra computational overhead while achieving competitive or better performance in most of the games.

Table 5: The running time of **one iteration** of different methods in different games (second).

| Game | KL | EU | ME | ML | UMD (RS) | UMD (GLD) |
|---|---|---|---|---|---|---|
| Kuhn_A | 0.0298 | 0.0301 | 0.0298 | 0.0313 | 0.0299 | 0.0309 |
| Kuhn_B | 0.0292 | 0.0311 | 0.0299 | 0.0325 | 0.0303 | 0.0309 |
| Leduc_A | 0.1623 | 0.1630 | 0.1631 | 0.1664 | 0.1988 | 0.1993 |
| Leduc_B | 0.1817 | 0.1861 | 0.1850 | 0.1839 | 0.2178 | 0.2173 |
| Tiny_Hanabi_A | 0.0271 | 0.0265 | 0.0262 | 0.0276 | 0.0280 | 0.0273 |
| Tiny_Hanabi_B | 0.0257 | 0.0261 | 0.0255 | 0.0269 | 0.0267 | 0.0263 |
| Tiny_Hanabi_C | 0.0261 | 0.0263 | 0.0253 | 0.0272 | 0.0269 | 0.0259 |
| Tiny_Hanabi_D | 0.0259 | 0.0266 | 0.0258 | 0.0265 | 0.0274 | 0.0258 |
| Kuhn | 0.0200 | 0.0203 | 0.0199 | 0.0196 | 0.0239 | 0.0233 |
| Leduc | 0.2692 | 0.2701 | 0.2658 | 0.2695 | 0.3445 | 0.3429 |
| Goofspiel | 0.0167 | 0.0167 | 0.0166 | 0.0166 | 0.0191 | 0.0189 |
| Liars_Dice | 0.2454 | 0.2284 | 0.2258 | 0.2228 | 0.3057 | 0.3302 |
| Hex | 0.0029 | 0.0029 | 0.0027 | 0.0027 | 0.0051 | 0.0051 |
| Dark_Hex | 0.0149 | 0.0153 | 0.0150 | 0.0149 | 0.0225 | 0.0227 |
| Blotto | 0.0154 | 0.0159 | 0.0158 | 0.0156 | 0.0162 | 0.0162 |
| Quoridor | 0.0016 | 0.0016 | 0.0016 | 0.0016 | 0.0028 | 0.0029 |
| Bargaining | 0.0785 | 0.0784 | 0.0778 | 0.0773 | 0.1026 | 0.1031 |
| Auction | 0.1467 | 0.1501 | 0.1480 | 0.1464 | 0.1484 | 0.1515 |
| Oh_Hell | 0.1448 | 0.1448 | 0.1440 | 0.1427 | 0.2272 | 0.2319 |
| Trade_Comm | 0.0082 | 0.0082 | 0.0082 | 0.0081 | 0.0100 | 0.0103 |
| MCC_Kuhn_A | 0.6381 | 0.6497 | 0.6493 | 0.6380 | 0.6385 | 0.6451 |
| MCC_Kuhn_B | 0.6773 | 0.6707 | 0.6715 | 0.6645 | 0.6604 | 0.6625 |
| MCC_Goofspiel_A | 0.5186 | 0.5123 | 0.5094 | 0.5029 | 0.5012 | 0.5042 |
| MCC_Goofspiel_B | 0.5172 | 0.5361 | 0.5295 | 0.5238 | 0.5246 | 0.5085 |

