# OpenReview forum: "Unified Mirror Descent: Towards a Big Unification of Decision Making"
_ICLR.cc/2024/Conference — Submitted to ICLR 2024_

### Official Review · Reviewer_5a1i · 2023-10-26

**Soundness:** 2 fair
**Presentation:** 1 poor
**Contribution:** 2 fair
**Rating:** 3
**Confidence:** 4

**Summary:**

The submission proposes a meta algorithm for combining different mirror descent and mirror descent-like update rules that they call UMD. The submission evaluates UMD across single-agent settings, zero-sum settings, cooperative settings, and general-sum settings.

**Strengths:**

Originality: I am not aware of other meta algorithms for combining mirror descent-like update rules in decision-making settings.

Quality: The submission does a thorough job of comparing against baselines in tabular settings.

**Weaknesses:**

### Comments I made while reading

> As one of the most popular algorithms, mirror descent (MD) (Vural et al., 2022)

It's a bit strange to cite a paper from 2022 as the main reference for mirror descent.

---

> A decision-making problem, either single-agent, cooperative multi-agent, competitive multi-agent,
or mixed cooperative-competitive settings, can be described as a decentralized partially observable Markov decision process (Dec-POMDP) (Oliehoek & Amato, 2016)

This is **not true**. A Dec-POMDP is necessarily fully cooperative (see Oliehoek & Amato, 2016). A better formalism to encompass the problem settings the submission cares about is partially observable stochastic games (POSGs).

---

In the problem statement section, the submission includes the idea that it is considering soft-optimal policies (as opposed to optimal) as a footnote. This is a big difference and ought to be defined rigorously and not merely listed as a footnote.

Furthermore, the submission's presentation of its solution concept has some issues. First, it is actually unclear what solution concept the submission is using because H(pi_i) is not defined. Does it mean H(pi_i(\tau_i^t)) or does it mean the expected entropy accumulated by pi_i? Second, the submission claims that this solution concept is a Nash equilibrium. Yet, given the presence of entropy regularization, it is clear not a Nash equilibrium. If it H(pi) = H(pi_i(\tau_i^t)), then it is an agent quantal response equilibrium, which is a well-established solution concept.

---

> In other cases, the evaluation metric for a joint policy is the distance of the policy to the NE, called the NE-Gap

In zero-sum games, there's an obvious justification for this evaluation metric. However, in general-sum games it's less clear that NE-Gap is meaningful. At the very least, this merits some discussion.

---

> little has been known for more complicated cases including general-sum and mixed cooperative-competitive settings.

This isn't exactly right. It's known that computing equilibria in these settings is computationally hard. Yet, there are still many works studying the settings. Perhaps most significantly (at least on the empirical side), is recent work on Diplomacy (specifically, Mastering the Game of No-Press Diplomacy via Human-Regularized Reinforcement Learning and Planning). However, note that this line of research neither uses Nash equilibria nor quantal response equilibria as the solution concept.

---

In section 4, the solution concept of interest becomes clear via equation (2). (Specifically, that the submission is studying quantal response equilibria.)

---

> In single-agent and two-player zero-sum (i.e., purely competitive) settings, the most commonly used method to solve the problem (2) is mirror descent.

This is a big claim and I don't think it's true. At least most single-agent RL papers do not use mirror descent. For two-player zero-sum games, other methods based on homotopies are also common.

---

> f is called the regularizer

I'm not sure what this means. In mirror descent, f in equation (3) would be the Bregman divergence. Is the submission just using a different name for Bregman divergence (if so, why?)?

---

In section 4.1, it becomes clear that the submission does not use f to mean a Bregman divergence. **This is an issue** because it means that equation (3) is not necessarily a mirror descent update rule.

---

All of the material from the start of Section 4.1 up through equation (8) can be found in (Sokota et al., 2023). Using this material without clear attribution is sketchy, at best. The right thing to do here would be to include this material as background, not in the contribution section.

---

After the derivation of (8), the submission discusses a seemingly ad hoc projection. This doesn't make a lot of sense. The required projection for (magnetic) mirror descent is dictated by the Bregman divergence. It is not truly an instance of (magnetic) mirror descent with the normalization scheme suggested by the submission.

---

There ought to be a new paragraph after the period here: "with the Euclidean distance. In addition,"

---

I'm a bit unclear on equation (9). Is phi part of the summation? If not, then this is a special case of the optimization problem that was already stated? If so, then this isn't computing an agent quantal response equilibria.

---

In the RS algorithm, what exactly is L in each setting?

---

Given the lack of clarity about the solution concept, the experimental results are also unclear. As best I can tell, the submission is using agent quantal response equilibria. However, agent quantal response equilibria are not solutions to minimax problems, so I do not know what the submission means by "gap" in Figure 4.

---

> (iv) For the four baselines, none of them can consistently outperform all the others across different types of games, which supports the motivation of this work

How did the submission select the hyperparameters for each of the algorithms. This plays an important role in the results. Also, how much regularization is the submission using? This also plays an important role. The claim above is unjustified without details on these issues and without clarification on what the y-axis is measuring.

---

For the tiny Hanabi results, I would be interested to see performance on games E and F as well.

---

Regarding the summarizing results in Figure 1, it seems like the normalized improvement of UMD over KL is at most small and sometimes significantly less than 1 (such as in Leduc). This raises some questions as to whether it's worth the effort.

---

Are all of the results full feedback?

---

Is the submission is also using the same magnet update scheme for the Euclidean version? That would be incorrect -- the Euclidean version needs to update the magnet with a Euclidean update (i.e., $\rho_{k+1} = (1 - \hat{\eta}) \rho_k + \hat{\eta} \pi_{k+1}$).

---

One thing I'm a bit confused about is that it seems like the update equations are solving for different for different things. KL and EU with moving magnets are trying to solve for Nash. But ME and ML are trying to solve for phi regularized equilibria? Does the submission actually tell the reader what phi ME and ML are using?

---

### Summary

There are a variety of basic issues that make it hard to seriously consider the submission for acceptance. Some are summarized below:
- The submission incorrectly defines fundamental terms Dec-POMDP and Nash equilibria.
- The material in 4.1 up through equation 10 is background material, not the contribution.
- The "practical version" of equation 8 is (for unclear reasons) not even an instance of (magnetic) mirror descent.
- The evaluation metric is unclear. The submission seems like it's actually using agent quantal response equilibria, but there's no such thing as an AQRE gap.
- It is unclear based on the description in the text whether the candidate algorithms are even solving for the same solution concept.

**Questions:**

> Think of the things where a response from the author can change your opinion, clarify a confusion or address a limitation.

There are a host of issues discussed above that require significant revisions to the paper. I would possibly change my opinion if the authors posted a revised version in which all of the issues were addressed.

---

### Official Review · Reviewer_crN4 · 2023-10-27

**Soundness:** 2 fair
**Presentation:** 3 good
**Contribution:** 2 fair
**Rating:** 5
**Confidence:** 4

**Summary:**

The paper proposes a unified approach for addressing sequential decision-making problems that mixes well-known base strategies with the intent of performing best on all possible types of single and multiagent environments.
The proposed method uses scalar weights to mix 4 well-known base policies and optimizes such weights based on an evaluation metric (expected reward or Nash-Equilibrium gap) using zeroth-order gradient approximations.
Experiments are performed on tabular environments demonstrating the superiorness of such an approach compared to using the base policies.

**Strengths:**

- The paper is well motivated by the fact that no single best algorithm exists for the considered sequential decision-making problems. Even among similar environments (i.e. multi-agent mixed cooperative-competitive), none of the base policies perform best on all.
- Experiments are extensive and different analyses are performed to illustrate the goodness of the approach

**Weaknesses:**

- The idea is quite simple and in my opinion it only partially solves the underlying problem.
- To update the weights, it is required to compute expected rewards or NE gaps (in multiagent environments) for different perturbations. If I am not mistaken, this requires full knowledge of the game payoffs unlike the base policies which only get updated based on observed rewards. Moreover, computing NE gaps also requires knowing policies of the other agents.
- Experiments are only tabular and of course it would be interesting to know how the approach scales and performs with neural network policies.

**Questions:**

- In Figure 7, I understand that vanilla RS and vanilla GLD are slow in up/down weighting the base policies. But, what is the explanation for them assigning higher weight to EU in the long run?

- In the authors' view, is it really required to perform zeroth-order optimization by estimating gradients using the proposed methods, or a bandit (e.g. EXP3) strategy could also perform well? In my opinion this is the main bottleneck of the approach and it would be good to understand whether such extra computation and knowledge (to estimate the gradients) is indeed necessary.

---

### Official Review · Reviewer_amuq · 2023-10-27

**Soundness:** 2 fair
**Presentation:** 1 poor
**Contribution:** 2 fair
**Rating:** 5
**Confidence:** 3

**Summary:**

The papers considers Decentralized POMDP framwork from (Oliehoek &
Amato, 2016) which contains as special cases: MDP, RL, Stochastic
games with possibly partial observations, and possibly non zero-sum.
The discounted cumulative payoff is considered. The goal of the paper
is to propose in the tabular setting a single "algorithm", which, if
used by each player, perform effectively in the various problems
contained in the framework. No theoretical guarantee is provided but
extensive numerical experiments are presented.

I believe there are several issues that need the authors attention. I will
be happy to raise my rating if these are addressed.

**Strengths:**

na

**Weaknesses:**

I wish the authors recalled the motivation for considering soft
optimality rather than plain optimality.

One issue with the present submission is that the formalism is
somewhat inconsistent and confusing:
- Shouldn't it be $\mathcal{T}_i=\bigcup_{t\geqslant 0}^{}(\mathcal{O}_i\times \mathcal{A}_i)^t$ ?
- The formalization of a policy is weird, shouldn't it be
  $\pi_i:\mathcal{T}_i\to \Delta(\mathcal{A}_i)$ ?
- The definition of (soft) NE as written in (1) does not make sense to
  me. The argmax seems to be taken on all policies $\pi_i$ for player
  $i$, as this policy is then used in the action-value function. But,
  it is also written that $\pi_i\in \Pi_i=\Delta(\mathcal{A}_i)$, but
  this is not the set of all policies, but merely the set of mixed
  actions. Moreover, if $\pi_i$ is indeed a policy, meaning a map
  $\pi_i:\mathcal{T}_i\to \Delta(\mathcal{A}_i)$, I don't know what
  the definition of the Shannon entropy $\mathcal{H}(\pi_i)$ is. Of
  course, I would understand what it meant if $\pi_i$ were an element
  of the simplex $\Delta(\mathcal{A}_i)$.

The effort from the authors to cite recent related research is
noteworthy. However, they don't really mention important historical
references. For instance, the first mention of MD should be
accompanied by the citation of the historical works [2-3].
Stochastic games were initially introduced by [4], and the prevalent
approach for decades was the uniform approach [5-6]. And so on.

An awkward issue is that the name (and acronym) Unified Mirror Descent
(UMD) already exists and designates something quite different in the
work [1]. The authors should pick an alternative name to avoid
confusion.

[1] Juditsky, A., Kwon, J., & Moulines, É. (2023). Unifying mirror descent and dual averaging. Mathematical Programming, 199(1-2), 793-830.

[2] Nemirovsky, A. S., & Yudin, D. B. (1983) Problem complexity and method efficiency in optimization. Wiley-Interscience Series in Discrete Mathematics.

[3] Beck, A., & Teboulle, M. (2003). Mirror descent and nonlinear projected subgradient methods for convex optimization. Operations Research Letters, 31(3), 167-175.

[4] Shapley, L. S. (1953). Stochastic games. Proceedings of the national academy of sciences, 39(10), 1095-1100.

[5] Mertens, J. F., & Neyman, A. (1981). Stochastic games. International Journal of Game Theory, 10, 53-66.

[6] Mertens, J. F., Sorin, S., & Zamir, S. (2015). Repeated games (Vol. 55). Cambridge University Press.

**Questions:**

na

---

### Official Review · Reviewer_4eSm · 2023-10-28

**Soundness:** 1 poor
**Presentation:** 1 poor
**Contribution:** 2 fair
**Rating:** 3
**Confidence:** 3

**Summary:**

This paper considers general reinforcement learning problems including both single-agent and multi-agent scenarios. It proposes unified mirror descent (UMD) algorithm, which convexly combines four mirror descent-based update rules and uses zeroth-order optimization methods to find the combination weights. It conducts extensive experiments on 24 benchmark environments and shows that UMD can perform comparably or better than all other baselines in most benchmark environments.

**Strengths:**

The experiments on tabular environments are extensive and demonstrate the advantage of the proposed RS/GLD methods over the vanilla RS/GLD methods.

**Weaknesses:**

One major weakness of this paper is that it provides very little justification of the proposed algorithms. For example, it is not clear that "*UMD could inherit the properties of these algorithms*" even from an intuitive perspective. From my perspective, **UMD** implicitly assumes that the optimal solution set of the problem is convex since if it is not convex and the four different update rules converge to different optimal solutions, then a convex combination of them can easily result a suboptimal solution. Is that possible to run experiments on an environment with non-convex optimal solution set? From this point of view, running experiments using neural networks-based policy on more complicated environments is critical since they have more complicated optimal solution set.

Another weakness is that **UMD** looks more like an algorithm selection strategy than an algorithm framework, if my understanding is correct. In particular, based on the experiment results in Figure 1, although **UMD** outperforms or matches the baselines in many environments, it outperforms all baselines in only a few settings. That is, a naive strategy such as "running all four update rules and picking the best one" will perform comparably with **UMD** without consuming more computational resources. From this point of view, it can be hard to see the advantage of using **UMD**.

### Suggestions on Writing
- Figure 1 can be put at a better place. For now, it is first mentioned in text on page 7 but placed on page 2.
- The presentation clarity of section 4.1 can be significantly improved. For example, should we add all $\pi_k$'s in the RHS of Eq. (6), (8) and (10) with corresponding superscripts "KL", "EU", "ME" and "ML"? If not, $\pi_{k+1}$'s in the LHS of Eq. (5) and (7) should be added with corresponding superscripts. Other clarification questions are given in **Questions** section.

**Questions:**

- What role does the magnet policy $\rho$ play conceptually in the update rule (4) and why it should follow the update rule $\rho_{k+1}(a)\propto\rho_k(a)^{1-\hat{\eta}}\pi_{k+1}(a)^{\hat{\eta}}$? What is the definition of $\hat{\eta}$? Is this magnet policy update rule the same for both "KL" and "EU" or does "EU" have different update rule for $\rho$? Or do update rules of both $\pi^{\mathrm{KL}}$ and $\pi^{\mathrm{EU}}$ use the same magnet policy $\rho$?
- Where is the part of the policy update rules that utilizes Eq. (9)?
- Does $v_k(a)$ denotes a single-step reward or the cumulataive reward? Can you define it mathematically?
- Is the objective $\mathcal{L}(\alpha)$ defined with respect to the current policy? If yes, then it looks $\mathcal{L}(\alpha)$ is a constantly changing objective. Then, why can the proposed hyperparameter optimization method still work?
- It seems the update magnitude of $\alpha$ is determined by purely random sampling. Why can this random sampling work?

---

### Official Review · Reviewer_Wqtu · 2023-10-30

**Soundness:** 3 good
**Presentation:** 3 good
**Contribution:** 2 fair
**Rating:** 5
**Confidence:** 3

**Summary:**

## General
This paper proposed a unified method of policy iteration which works in 5 different types of RL problems: single-agent, competitive zero-sum, competitive general-sum, cooperative, mixture of cooperative-competitive. The algorithm works in a Decentralized-POMDP setup and look for a policy in optimizing value function in single / cooperative case, and approaching Nash-Equilibrium otherwise.

## Construction
- The key construction of this algorithm is, at each moment (time) and for each agent, the convex combination of 4 external  basepolicy updates from *Mirror descent*: limiting update step via KL / Euclidean, and Linear / Exponential Multiplicative Weight Updates. The construction of the 4 external policies contains the processing of the whole Dec-POMDP environment, which are inherited by this algorithm, thus the construction can be reduced to deal with only one agent at each particular time.

- In the above construction, only the coefficients $\alpha$ of the convex combination is modifiable. The rest of the algorithm is to optimize $\alpha$ gradually. Two options of updating $\alpha$ is given:
    - Random Search
    - GradientLess Descent

Assembling the above 2 parts, one can find a policy in updating 4 policies together with the coefficients in a same updating sequence.

## Experiments
24 environments are given in the above 5 types of games. They can converge in iterations following the best one of the 4 base policy.

**Strengths:**

- The algorithm is simple and is built on other solid methods.

- The algorithm works in solving the problems listed in the paper.

- The algorithm does not provide too much additional computational complexity except the sum of the 4 external base algorithms.

**Weaknesses:**

- The novelty of the paper is relatively weak: as far as I can see, it is only combining existing 4 algorithms via convex combination.

- I would hope to see some more theoretical analysis about why this method work, and what really dominates the limit of $\alpha$.

- Maybe discussing the roles of the 4 different base policies in all 5 kinds of setups could help understanding the unified methods.

**Questions:**

Questions are sorted by the time they show up, not by their importance.

1. By using convex combination of the updating steps, or roughly say *gradients*, it is equivalent to optimize the same convex combination of the external base objective functions. The authors probably could analyse the simplex $\Delta$ (of dimension 3) of all convex combinations of the objective functions, and state the effectiveness and generalisability of the method in this way.

2. Following Question 1, why is the **optimal policy** occurs within the simplex and what does the optimal policy, as the convex combination of 4 existing policies, really mean?

3. By combining the result policies of the 4 algorithms, is there a problem which could not be solved by any of them but the UMD method can solve?

4. Minor: On page 5, when introducing ME and ML updating methods, I did not find the definition of $v_k$ and $\phi$.

5. In this unified framework, is there any evidence of the (possible) completeness of base methods, even for a single kind of methods? Or in other words, could there be a choice of complete methods $\mathcal{M}(P)$ for problem $P$, such that adding any other methods (equivalently, adding additional objective function) would result in that the components of $\alpha$ outside the complete set $\mathcal{M}(P)$ would always converge to 0.

---

### Meta-Review · Area_Chair_FSez · 2023-12-05

**Metareview:**

The papers proposes a unified learning algorithm in the context of decentralized partially observable Markov decision processes (POMDPs) which includes as special cases standard MDPs, (tabular) reinforcement learning, and stochastic games (possibly partially observable). The main contribution of the paper is a "template-like" algorithmic scheme which, if employed by all players, is claimed to achieve good results in the various frameworks under consideration. This claim is not substantiated with theoretical results, but the authors perform an extensive series of numerical experiments to quantify this claim.

The reviewers identified several points where the paper could be improved - mainly regarding the lack of concrete theoretical take-aways. The authors did not provide a rebuttal to the reviewers' input, so a decision was reached to make a "reject" recommendation.

**Justification For Why Not Higher Score:**

No rebuttal to the reviewers' (valid) criticisms.

**Justification For Why Not Lower Score:**

N/A

---

### Decision · Program_Chairs · 2024-01-16

Reject